# UC-NeRF: Neural Radiance Field for Under-Calibrated Multi-view Cameras in Autonomous Driving

**Kai Cheng**[1*], **Xiaoxiao Long**[2,3*], **Wei Yin**[4], **Jin Wang**[1], **Zhiqiang Wu**[3], **Yuexin Ma**[5], **Kaixuan Wang**[4], **Xiaozhi Chen**[4], **Xuejin Chen**[1†]

[1] MoE Key Laboratory of Brain-inspired Intelligent Perception and Cognition, University of Science and Technology of China  [2] The University of Hong Kong
[3] PKU-Wuhan Institute for Artificial Intelligence  [4] DJI Technology  [5] ShanghaiTech University

## Abstract

Multi-camera setups find widespread use across various applications, such as autonomous driving, as they greatly expand sensing capabilities. Despite the fast development of Neural radiance field (NeRF) techniques and their wide applications in indoor and outdoor scenes, applying NeRF to multi-camera systems remains very challenging. This is primarily due to the inherent under-calibration issues in multi-camera setup, including inconsistent imaging effects stemming from separately calibrated image signal processing units in diverse cameras, and system errors arising from mechanical vibrations during driving that affect relative camera poses. In this paper, we present UC-NeRF, a novel method tailored for novel view synthesis in under-calibrated multi-view camera systems. Firstly, we propose layer-based color correction to rectify the color inconsistency in different image regions. Second, we propose virtual warping to generate more viewpoint-diverse but color-consistent virtual views for color correction and 3D recovery. Finally, spatiotemporally constrained pose refinement is designed for more robust and accurate pose calibration in multi-camera systems. Our method not only achieves the state-of-the-art performance of novel view synthesis in multi-camera setups but also effectively facilitates depth estimation in large-scale outdoor scenes with the synthesized novel views.

## 1 Introduction

Neural radiance field (NeRF) is a revolutionary approach that enables the synthesis of highly detailed and photorealistic 3D scenes from 2D images. This technology has opened up a multitude of new possibilities in autonomous driving, such as generating synthetic data from diverse viewpoints for robust training of perception models and providing effective 3D scene representations to enhance comprehensive environmental understanding (Fu et al. (2022); Zhang et al. (2023)).

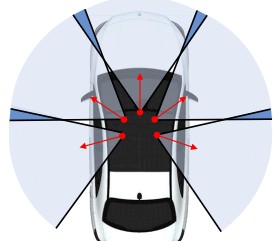

Figure 1: Illustration of a multi-camera system in autonomous driving.

Multi-camera systems (Sun et al. (2020); Caesar et al. (2020); Guizilini et al. (2020)) are commonly used for autonomous driving, while involving the strategic placement of multiple cameras to capture a holistic perspective of the surrounding environment, as shown in Fig. 1, supplying spatially consistent information to complement the temporal data for perception tasks (Mei et al. (2022); Pang et al. (2023)). Incorporating NeRF in multi-camera systems could provide a way to efficiently and economically produce extensive high-quality video data for training various models in autonomous driving systems.

However, naively combining images captured from multi-camera systems into NeRF's training often results in a significant deterioration of rendering quality, as depicted in the first two rows of

---

*First two authors contributed equally.
†Corresponding athuor (xjchen99@ustc.edu.cn).

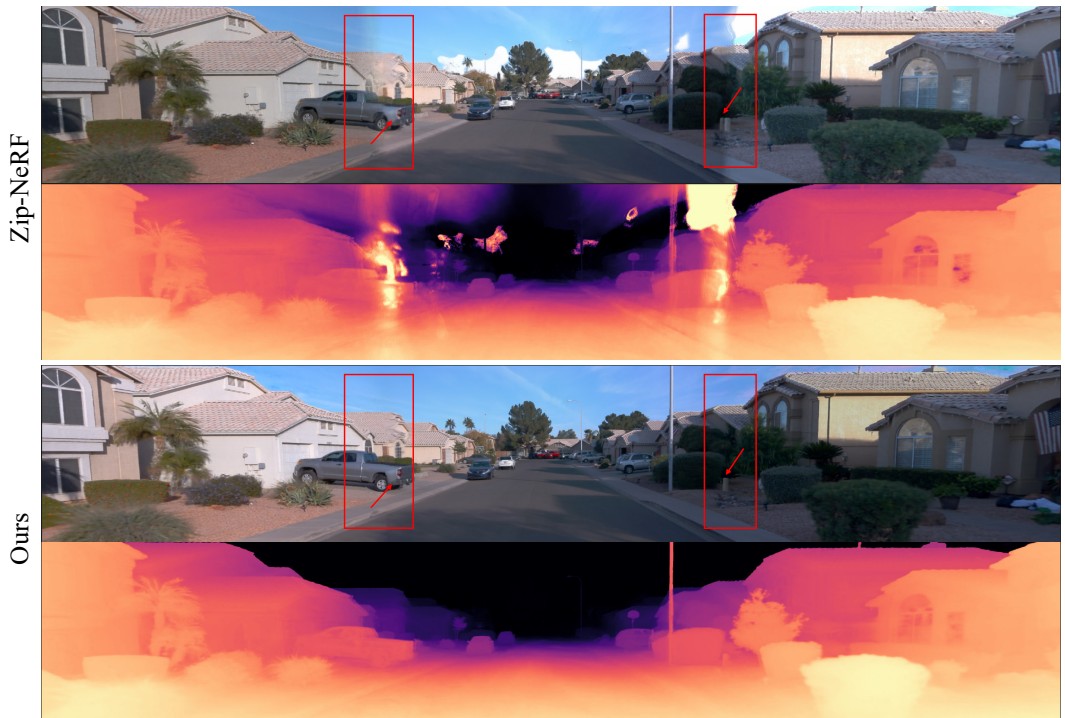

Figure 2: In under-calibrated multi-camera systems, the NeRF quality significantly degrades (the first row), along with color discrepancies (red boxes), object ghosts (red arrows), and wrong geometry (the second row). Our UC-NeRF achieves high-quality rendering and accurate geometry in the challenging cases.

Fig. 2. The underlying cause of this degradation is the inconsistent color supervision from different views, since the multi-cameras are usually under-calibrated. This under-calibration issue manifests in two distinct ways. First, the image signal processing (ISP) units involve various techniques like white balance correction, gamma correction, etc., to convert the raw data captured by the sensors to discretized pixel colors. However, these ISP units fluctuate over time across different cameras, resulting in color disparities within the same scene region, as shown in the red box in Fig. 2. Secondly, even with delicate camera pose calibration beforehand, systematic errors during vehicle production and vibrations during driving inevitably introduce misalignment and further exacerbate color inconsistency, as indicated by the arrows in Fig. 2.

Several NeRF methods (Martin-Brualla et al. (2021); Rematas et al. (2022); Tancik et al. (2022)) attempt to alleviate the inconsistent color supervision by modeling image-dependent appearance with a global latent code for each image. However, the capacity of a global latent code to uniformly correct colors in different regions of an image is limited, especially when different regions correspond to different color transformations. Furthermore, learning one color correction code for each image usually leads to overfitting when the training images lack color and viewpoint diversity. This limitation is pronounced for areas observed by side-view cameras, which have fewer observations and limited overlapping with front-view areas. To correct inaccurate poses, some approaches perform joint NeRF optimization with pose refinement using photometric losses. Unfortunately, utilizing such joint optimization under a multi-camera setup, the photometric consistency across cameras can not be ensured and spatial relations among cameras are not fully utilized, making optimization more challenging and prone to local minima.

To address these challenges, we introduce UC-NeRF, a method for high-quality neural rendering with multiple under-calibrated cameras. We introduce three key innovations: 1) **Layer-based Color Correction**. To address color inconsistencies in the training images, especially for those taken by different cameras, we design a novel layer-based color correction module. This module separately adjusts the colors of the foreground and sky regions using two learned affine transformations for each image. 2) **Virtual Warping**. We introduce a "virtual warping" strategy that generates viewpoint-

diverse yet color-consistent observations for each camera at each moment. These warped images under virtual viewpoints offer stronger constraints on the latent codes for color correction, especially for multi-camera systems where the overlapping region between cameras is limited. Moreover, the virtual warping strategy naturally expands the range of the training views for NeRF, enhancing its effectiveness in learning both the scene's appearance and geometry. 3) **Spatiotemporally Constrained Pose Refinement**. We propose a spatiotemporally constrained pose optimization strategy that explicitly models the spatial and temporal constraints between cameras for pose optimization. This approach also improves the robustness against photometric differences by utilizing reprojection errors during pose optimization.

Experiments on the public datasets Waymo (Sun et al. (2020) and NuScenes (Caesar et al. (2020)) show that our method achieves high-quality renderings with a multi-camera system and outperforms other baselines by a large margin. Moreover, we show that the obtained high-quality renderings of novel views can facilitate downstream perception tasks like depth estimation.

## 2 RELATED WORK

**Multi-view Stereo** Multi-view stereo (MVS) is a fundamental 3D vision task that aims to reconstruct a 3D model from posed images. Traditional methods (Campbell et al. (2008); Furukawa & Ponce (2009); Bleyer et al. (2011); Furukawa et al. (2015); Schönberger et al. (2016)) exploit pixel correspondences between images from hand-crafted features to infer 3D structure. Deep learning methods (Yao et al. (2018); Vakalopoulou et al. (2018); Long et al. (2020); Chen et al. (2019); Long et al. (2021); Ma et al. (2022); Feng et al. (2023)) generally build multi-view correspondences implicitly and regress the depth maps or 3D volumes in an end-to-end framework. Despite the increasing capability of MVS techniques in reconstructing accurate 3D models, it is strenuous to integrate 3D models into the traditional rendering pipeline to achieve photorealistic rendering.

**NeRF for Outdoor Scenes** NeRF (Mildenhall et al. (2021)) is a revolutionary technique that allows realistic novel view rendering without explicitly reconstructing 3D models. Its effectiveness in high-quality novel view synthesis has been demonstrated on indoor scenes and small-scale outdoor scenes. But it faces challenges when applied to unbounded outdoor scenes due to infinite depth range, complex illumination, and dynamic objects. To make NeRF more effective for infinite depth range, NeRF++ (Zhang et al. (2020)) divides the scene into foreground and background regions with an inverted sphere parameterization. In the following works (Barron et al. (2022); Wang et al. (2023a)), more complicated non-linear scene parameterization is proposed to model the outdoor space more compactly and sample points more efficiently. Some other works (Deng et al. (2022); Xie et al. (2023); Wang et al. (2023b); Yang et al. (2023)) learn the complex geometry of outdoor scenes by introducing depth and surface normal priors. Moreover, to adapt to the view-dependent appearance due to surface reflection, camera parameters, and environment change, several works (Martin-Brualla et al. (2021); Rematas et al. (2022); Tancik et al. (2022); Turki et al. (2022); Li et al. (2023)) learn appearance-related latent codes independently to control the view-dependent effect. Besides, some works (Xie et al. (2023); Turki et al. (2023)) model dynamic objects separately based on semantic priors, using 3D detection or semantic segmentation. In comparison, we primarily address the rendering quality deterioration problem caused by under-calibration of photometry and poses in a multi-camera setup for large-scale outdoor scenes.

**NeRF with Pose Refinement** NeRF methods always require accurate camera poses to optimize a neural 3D scene. However, the camera poses obtained from Structure-from-Motion (SfM) usually contain subtle errors that could significantly degrade the quality of the reconstructed NeRF. NeRF−− (Wang et al. (2021)) jointly optimizes camera parameters with NeRF training via the photometric loss. However, the pose optimization struggles to achieve effective updates due to the increased non-linearity of NeRF arising from position encoding. BARF (Lin et al. (2021)) eliminates this negative impact with a coarse-to-fine training strategy. SiNeRF (Xia et al. (2022)) and GARF (Shi et al. (2022)) replace the positional encoding with different activation functions to reduce non-linearity while maintaining the same rendering quality. Besides, SCNeRF (Jeong et al. (2021)) and SPARF (Truong et al. (2023)) propose different geometric losses to improve the pose accuracy further. However, directly employing these methods in multi-camera systems will lead to a bottleneck, since the spatial relation between cameras is not taken into consideration. MC-NeRF ( Gao et al. (2023)) is a contemporaneous work on multi-camera systems, but it focuses on optimiz-

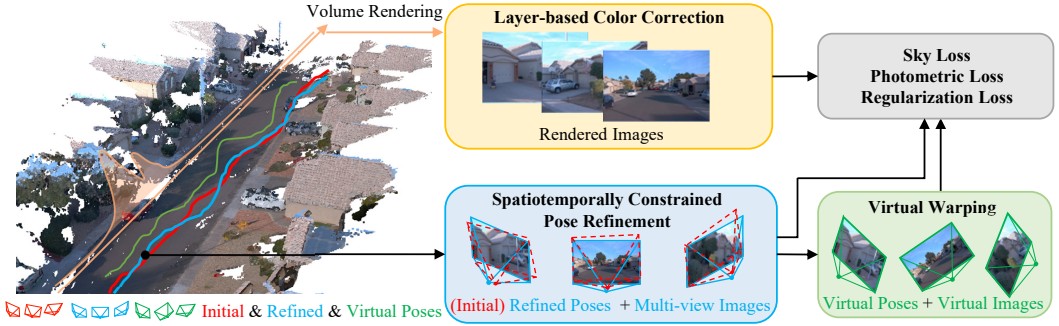

Figure 3: Overview of UC-NeRF framework. To mitigate the inconsistency of color supervision in multi-camera systems, the spatiotemporally constrained pose refinement module optimizes poses and the layer-based color correction module models the image-dependent appearance from varying cameras and timestamps. The virtual warping module generates diverse virtual views with geometric and color consistency, enriching data for color correction and 3D scene recovery.

ing the camera intrinsics during pose optimization while we propose to impose the spatiotemporal constraint between cameras for the pose optimization.

## 3 METHOD

Our UC-NeRF extends the general NeRF algorithm to the multi-camera setup in autonomous driving. We begin by reviewing the common NeRF pipeline. Then we introduce the layer-based color correction (Sec. 3.2) to reformulate the color rendering for handling the inconsistent color supervision in multi-camera systems. In Sec. 3.3, we introduce our virtual warping strategy to assist color correction by generating viewpoint-diverse but color-consistent images. Finally, the spatiotemporally constrained pose refinement is explained in Sec. 3.4.

### 3.1 PRELIMINARY

NeRF models a 3D scene as a continuous implicit function $\theta$ and regresses the density $\sigma$ and color $\mathbf{c} \in \mathbb{R}^3$ of every individual 3D point given its 3D coordinate $\mathbf{p} \in \mathbb{R}^3$ and a unit-norm viewing direction $\mathbf{d} \in \mathbb{R}^2$. To synthesize a 2D image, NeRF employs volume rendering which samples a sequence of 3D points along a camera ray $\mathbf{r}$ as $\mathbf{I}(\mathbf{r}) = \sum_{n=1}^{N} T_n \alpha_n \mathbf{c}_n$, where $T_n$ is the accumulated transmittance of the sampled points, $\alpha_n$ and $\mathbf{c}_n$ is the alpha value and the color of the sampled $n$-th point. Detailed definitions can be referred to NeRF (Mildenhall et al. (2021)).

To optimize a NeRF, the photometric loss between the rendered color $\mathbf{I}(\mathbf{r})$ and the ground truth color $\hat{\mathbf{I}}(\mathbf{r})$ from a set of sampled rays $\mathcal{R}$ is applied as:

$$\mathcal{L}_{\text{pho}}(\theta) = \sum_{\mathbf{r} \in \mathcal{R}} \left\| \hat{\mathbf{I}}(\mathbf{r}) - \mathbf{I}(\mathbf{r}) \right\|_2^2 . \tag{1}$$

### 3.2 LAYER-BASED COLOR CORRECTION

In multi-camera systems, different cameras always have distinct Image Signal Processor (ISP) configurations, resulting in inconsistent imaging colors for the same 3D region. As a result, optimizing a NeRF representation using such inconsistent images always causes low-quality renderings. The work Urban-NeRF (Rematas et al. (2022)) attempts to approximate a global linear compensation transformation for each view to alleviate the discrepancies of the views from different cameras. It is worth noticing that due to the non-linear property of the ISP process, it is insufficient to model the spatially varying color patterns using a single global compensation transformation. To balance the rendering quality and efficiency, we propose to split the scene into foreground-sky layers and model the color compensation transformation for each layer separately. This is because the sky regions are always much brighter than the foreground objects and they present distinct ISP imaging effects.

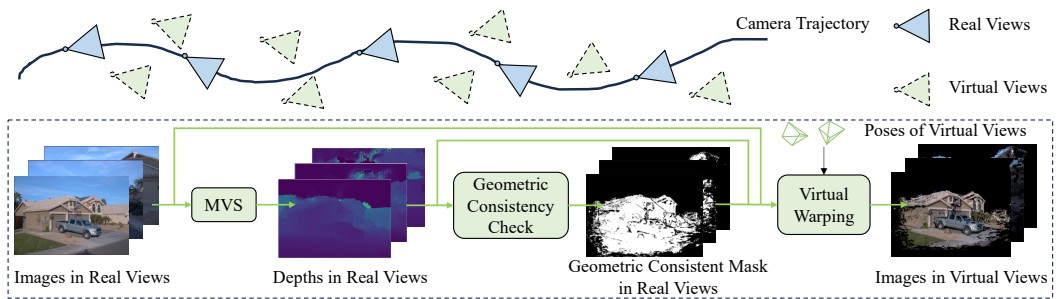

Figure 4: Generation of warped images in virtual views. For each known viewpoint, we estimate its depth map using a deep MVS method and filter out inaccurate depths through a geometric consistency check. Each virtual view is obtained by warping the image from a known viewpoint to the virtual viewpoint.

We first model the foreground and sky as two independent NeRF models $\theta_{fg}$ and $\theta_{sky}$. The color of a rendered pixel from ray $\mathbf{r}$ is obtained by the weighted combination of foreground color $\mathbf{I}_{fg}(\mathbf{r})$ and sky color $\mathbf{I}_{sky}(\mathbf{r})$, as illustrated in Eq. 2:

$$\mathbf{I}(\mathbf{r}) = \mathbf{I}_{fg}(\mathbf{r}) + (1 - o_{fg})\mathbf{I}_{sky}(\mathbf{r}), \tag{2}$$

where $o_{fg} = \sum_{n=1}^{N} T_{n,fg}\alpha_{n,fg}$ is the accumulated weight of foreground NeRF along $\mathbf{r}$. To encourage $o_{fg}$ to approach 1 in the foreground area and approach 0 in the sky area, a binary cross-entropy loss is employed as Eq. 3:

$$L_{sky}(o_{fg}, m_{sky}) = -m_{sky}\log(1 - o_{fg}) - (1 - m_{sky})\log(o_{fg}), \tag{3}$$

where $m_{sky}$ is the sky mask generated from pretrained segmentation model (Yin et al. (2022)).

After modeling the foreground and sky NeRF, we approximate the color correction of the foreground and the sky using separate affine transformations. Considering the color variance across both cameras and times, for each training image $\mathbf{I}_{i,k}$ from camera $k$ at time $i$, a foreground correction code and a sky correction code are assigned to represent the image-dependent color variation (*Subscripts $i, k$ are omitted in the following descriptions for clarity*). These correction codes are further decoded by a multi-layer perceptron (MLP) as the affine transformations $[\mathbf{A}, \mathbf{x}]$ and $[\mathbf{C}, \mathbf{y}]$, where $\mathbf{A}, \mathbf{C} \in \mathbb{R}^{3 \times 3}$, and $\mathbf{x}, \mathbf{y} \in \mathbb{R}^{3 \times 1}$. For the rendered pixel which emits $\mathbf{r}$ in $\mathbf{I}$, the final pixel color in Eq. 2 can be rewritten as:

$$\mathbf{I}(\mathbf{r}) = \mathbf{A}\mathbf{I}_{fg}(\mathbf{r}) + \mathbf{x} + (1 - o_{fg})(\mathbf{C}\mathbf{I}_{sky}(\mathbf{r}) + \mathbf{y}). \tag{4}$$

To stabilize the optimization process and ensure that the adjusted color does not significantly deviate from the original observation, we add a regularization term for the transformation matrices, as illustrated in Eq. 5:

$$L_{reg} = |\mathbf{A} - \mathbf{E}_3| + |\mathbf{C} - \mathbf{E}_3| + |\mathbf{x}| + |\mathbf{y}|, \tag{5}$$

where $\mathbf{E}_3$ refers to the identity matrix.

## 3.3 VIRTUAL WARPING

In multi-camera systems, images from different viewpoints often have limited overlapping areas, making it more challenging to align their colors compared to aligning frames from a single camera. To align the image colors of multiple cameras and prevent the optimized latent codes for color correction from overfitting to a specific view, we propose virtual warping, which simulates more diverse yet color-consistent images under a set of virtual viewpoints for training. Furthermore, virtual warping naturally expands the range of perspectives available to NeRF, thereby enhancing its capability to reconstruct the 3D scene. Fig. 4 shows the pipeline of our virtual warping. We employ a deep MVS method (Ma et al. (2022)) to estimate depth maps of all views. To remove outliers and retain the consistent depths across multiple views, we further leverage a geometric consistent check (Schönberger et al. (2016)) to generate a mask $\mathbf{M}$ that only keeps reliable depth values.

With estimated depths, we generate multiple virtual poses and warp colors and color correction codes to the virtual positions. Specifically, we perturb an existing pose $\mathbf{T}_o$ with an additional transformation $[\mathbf{R}_{o \to v}, \mathbf{t}_{o \to v}]$ as a virtual pose $\mathbf{T}_v$. The rotation matrix $\mathbf{R}_{o \to v}$ is generated by randomly selecting one of the three axes with a random angle $\in [-20°, 20°]$. The translation $\mathbf{t}_{o \to v}$ is a 3D vector of random direction with a length $\in [0m, 1m]$. Each pixel $\mathbf{p}_o$ in an existing image taken under camera pose $\mathbf{T}_o$ is warped to an image point $\mathbf{p}_v$ with the virtual pose $\mathbf{T}_v$ as:

$$d_v \overline{\mathbf{p}}_v = \mathbf{K}(\mathbf{R}_{o \to v} \mathbf{K}^{-1} d_o \overline{\mathbf{p}}_o + \mathbf{t}_{o \to v}), \tag{6}$$

where $\overline{\mathbf{p}}_o$ and $\overline{\mathbf{p}}_v$ is the homogeneous coordinates of $\mathbf{p}_o$ and $\mathbf{p}_v$, $\mathbf{K}$ is the camera intrinsic matrix, $d_v$ and $d_o$ are the pixel depth in the virtual view and the corresponding pixel depth in the original real view. Considering object occlusions, there could be multiple pixels in the original views mapped to the same position in the virtual view. We keep the warped pixel with the minimum depth.

After warping, the geometric consistency mask $\mathbf{M}$ is applied to the warped pixels to remove pixels with noisy depth. Then the color and the color correction code of the pixel in the original view are assigned to the corresponding warped pixels in the virtual view. This provides more clues to recover consistent appearance and geometry of the scene.

## 3.4 SPATIOTEMPORALLY CONSTRAINED POSE REFINEMENT

The rendering quality of NeRF heavily relies on the accuracy of camera poses. Previous approaches (Tancik et al. (2022); Xie et al. (2023)) model the camera poses independently and jointly optimize them within the NeRF framework. They do not fully exploit the spatial correlations between cameras in multi-camera systems, leading to under-constrained pose optimization. Additionally, camera pose optimization depends on the photometric consistency assumption, which is usually violated in long-time videos captured in multi-camera systems. Given the condition that the cameras have a fixed spatial relationship with the main capturing device (i. e. the driving car) during the whole process, we explicitly establish the temporally fixed geometric transformation between cameras.

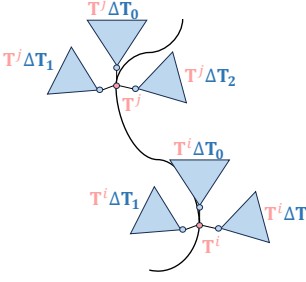

Figure 5: Pose modeling.

While capturing multi-view images with $K$ cameras, the pose $\mathbf{T}_k^i$ of the $k$th camera at time $i$ is denoted as the combination of the car's ego pose $\mathbf{T}^i$ and the relative transformation $\Delta\mathbf{T}_k$, which is temporally consistent and optimizable, as Fig. 5 shows. Explicitly modeling the spatial relationship between cameras provides more restrictions for pose refinement, thus effectively enhancing robustness against incorrect point matches across all frames.

After building the spatio-temporal constraint between camera poses, point correspondences between images captured by different cameras at different times are established as correlation graph $\mathcal{E}$. Then we employ bundle adjustment to minimize the reprojection error defined as:

$$L_{rpj} = \sum_{((i,k),(j,l)) \in \mathcal{E}} \left\| \mathbf{p}_l^j - \Pi_l \left( (\mathbf{T}^j \Delta\mathbf{T}_l)^{-1} \mathbf{T}^i \Delta\mathbf{T}_k \Pi_k^{-1} \left( \mathbf{q}_k^i \right) \right) \right\|^2, \tag{7}$$

where $\mathbf{p}_l^j$ and $\mathbf{q}_k^i$ are pixels in the images captured by camera $l$ at time $j$ and camera $k$ at time $i$, $\Pi_l$ and $\Pi_k^{-1}$ are projection function of camera $l$ and unprojection function of camera $k$.

## 3.5 TRAINING STRATEGY

Our NeRF training process consists of two stages. The first stage is pose refinement and depth estimation. We initialize the camera poses from sensor-fusion SLAM and further optimize them using our proposed spatiotemporally constrained pose refinement module, as described in Eq. 7. With these refined poses, we generate a depth map and geometric consistency mask for each image, following the procedure outlined in Sec. 3.3. The second stage is the end-to-end NeRF optimization. Specifically, the proposed layer-based color correction and virtual warping strategies are employed in the NeRF optimization to achieve high-quality renderings. In each training batch, we randomly sample $B$ real images and employ our virtual warping module to create $V$ virtual views for each real image. The pixels are randomly sampled from these real and virtual views as the ground truth

Table 1: Comparison on Waymo and NuScenes. Our method significantly outperforms other state-of-the-art methods on all the evaluation metrics.

| Method | Waymo | | | NuScenes | | |
|---|---|---|---|---|---|---|
| | PSNR ↑ | SSIM ↑ | LPIPS ↓ | PSNR ↑ | SSIM ↑ | LPIPS ↓ |
| Mip-NeRF (Barron et al. (2021)) | 22.42 | 0.698 | 0.471 | 23.31 | 0.758 | 0.489 |
| Mip-NeRF 360 (Barron et al. (2022)) | 24.46 | 0.769 | 0.406 | 25.15 | 0.809 | 0.436 |
| Instant-NGP (Müller et al. (2022)) | 23.84 | 0.702 | 0.494 | 23.81 | 0.777 | 0.476 |
| S-NeRF (Xie et al. (2023)) | 24.89 | 0.772 | 0.401 | 26.02 | 0.824 | 0.415 |
| Zip-NeRF (Barron et al. (2023)) | 26.21 | 0.815 | 0.389 | 27.06 | 0.831 | 0.435 |
| UC-NeRF (Ours) | **28.13** | **0.842** | **0.356** | **30.20** | **0.876** | **0.374** |

for NeRF training. Our UC-NeRF renders these pixels based on Eq. 4 and is supervised by the loss function in Eq. 8:

$$L = L_{pho} + \lambda L_{\text{sky}} + \gamma L_{\text{reg}}, \tag{8}$$

where $\lambda$ and $\gamma$ are the weights of $L_{\text{sky}}$ and $L_{\text{reg}}$.

## 4 EXPERIMENTS

### 4.1 DATASETS AND IMPLEMENTATION DETAILS

**Datasets.** We conduct experiments on two urban datasets with images captured with multi-camera settings, $i.e.$, Waymo (Sun et al. (2020)) and NuScenes (Caesar et al. (2020)). We select ten static scenes in Waymo and five static scenes in NuScenes for evaluation. To evaluate the performance of novel view synthesis, following common settings, we select one of every eight images of each camera as testing images and the remaining ones as training data. We apply the three widely-used metrics for evaluation, $i.e.$, peak signal-to-noise ratio (PSNR), structural similarity index measure (SSIM), and the learned perceptual image patch similarity (LPIPS) (Zhang et al. (2018)).

**Baselines.** We choose Zip-NeRF (Barron et al. (2023)) as our baseline using the codes implemented by Gu (2023). We compare our method with the baseline and other NeRF methods, including Mip-NeRF (Barron et al. (2021)), Mip-NeRF 360 ( Barron et al. (2022)), Instant-NGP (Müller et al. (2022)), and S-NeRF (Xie et al. (2023)). We provide implementation details in the appendix.

### 4.2 RESULTS OF NOVEL VIEW SYNTHESIS

The comparison of neural rendering results for urban scenes with multi-camera settings is shown in Tab. 1. With the proposed layer-based color correction, virtual warping, and spatio-temporally constrained pose refinement, our UC-NeRF outperforms the other methods on both datasets. We also show the panoramic rendering results in Fig. 6. Without any anti-aliasing design in the rendering process, images generated by Instant-NGP and S-NeRF exhibit notable blurriness. Although Zip-NeRF features an anti-aliasing mechanism, it also amplifies the artifacts caused by inconsistent color supervision across different views. Our approach excels at rendering consistent colors and sharp details, as highlighted in the regions of texts, cars, and buildings. Additionally, our method provides more accurate 3D reconstruction, as demonstrated by the depth maps. We show more results of the Waymo and NuScenes datasets in the appendix.

### 4.3 ABLATION STUDY

We conduct extensive ablation studies on ten scenes from the Waymo dataset to explore the effect of each proposed module in our UC-NeRF. We investigate the effect of each module, i.e., layer-based color correction (LCC), spatiotemporally constrained pose refinement (STPR), and virtual warping (VW). As shown in Tab. 2, the layer-based color correction module brings significant improvement (the third row)

Table 2: Ablation study.

| LCC | STPR | VW | PSNR ↑ | SSIM ↑ | LPIPS ↓ |
|---|---|---|---|---|---|
| ✗ | ✗ | ✗ | 26.21 | 0.815 | 0.389 |
| ✗ | ✓ | ✗ | 26.95 | 0.839 | 0.360 |
| ✓ | ✗ | ✗ | 27.18 | 0.820 | 0.375 |
| ✓ | ✗ | ✓ | 27.26 | 0.825 | 0.372 |
| ✓ | ✓ | ✗ | 27.82 | 0.838 | 0.371 |
| ✓ | ✓ | ✓ | **28.13** | **0.842** | **0.356** |

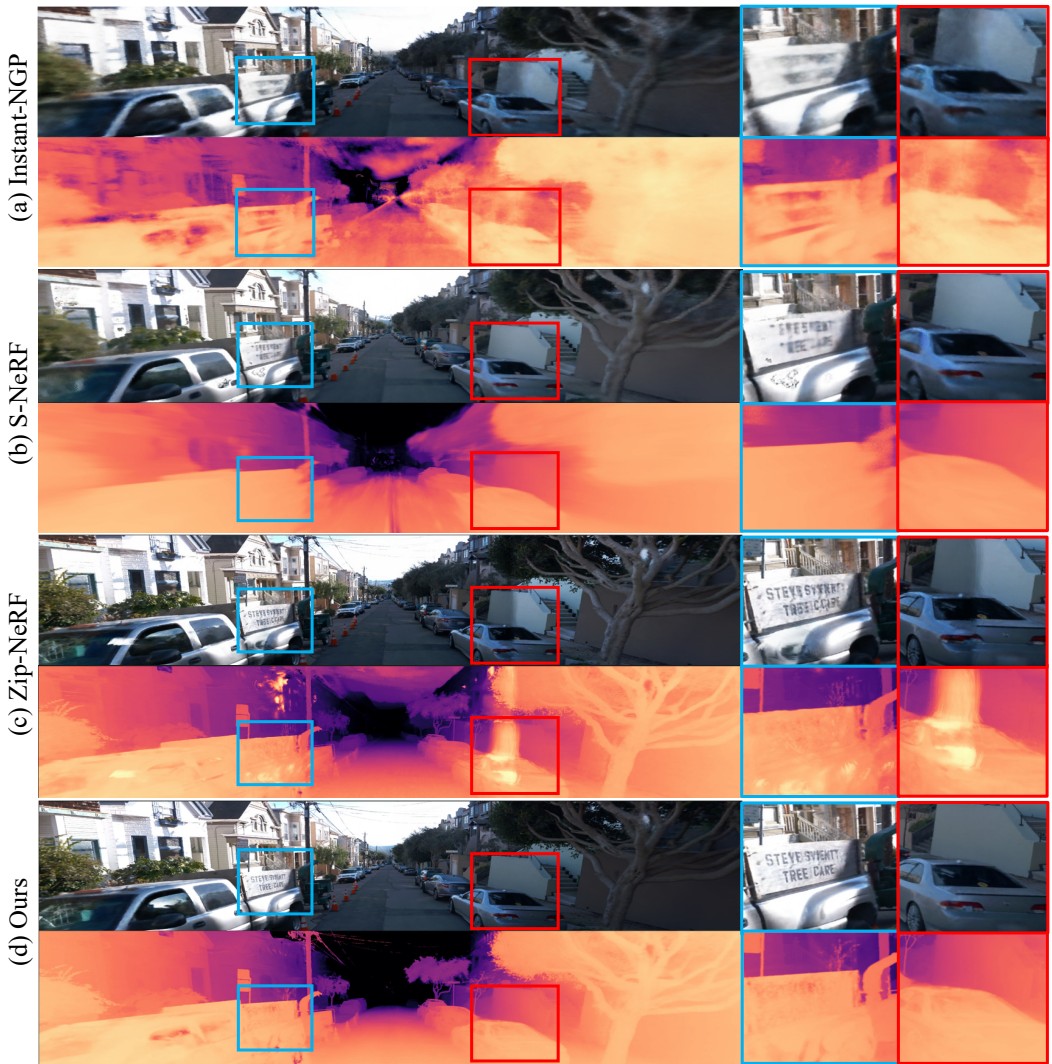

Figure 6: Comparison of rendered panoramas at a resolution of $5760 \times 1280$. Notable enhancements are indicated in the blue and red boxes. The close-ups of some patches are displayed to emphasize the rendering details. Compared to other methods, our results present consistent colors and sharp details, even faithfully recovering the slogans. Please refer to the appendix for additional results.

compared with the baseline model, since it solves the problem of inconsistent color supervision between views in training. Fig. 7 (b) also illustrates that the LCC module reduces hazy artifacts and presents sharper renderings. By incorporating the spatiotemporally constrained pose refinement (STPR) module, the quality of rendering is further improved. Moreover, our virtual warping (VW) strategy can enrich the diversity of training views for learning color correction, appearance, and geometry. Even the object details, e.g. the car lights and the car emblem, become more discernible in Fig. 7 (d). One notable thing is that the accuracy of the virtual views provided by virtual warping is closely related to the accuracy of the poses. Thus, virtual warping provides a more noticeable boost when the pose refinement module is added (the fourth and sixth row in Tab. 2). More detailed discussions can be found in the appendix.

**Benefit of Virtual Warping** Our virtual warping enriches NeRF's training perspectives of each camera by generating images with consistent geometry and appearance. In addition to the overall improvement in rendering quality shown in Tab. 2, we present more cases that demonstrate significant enhancement in rendering quality after incorporating virtual warping in Fig. 8. We can see better color correction (the first row) and enhanced image details (the second row).

(a) Baseline            (b) + Layer-based Color Correction

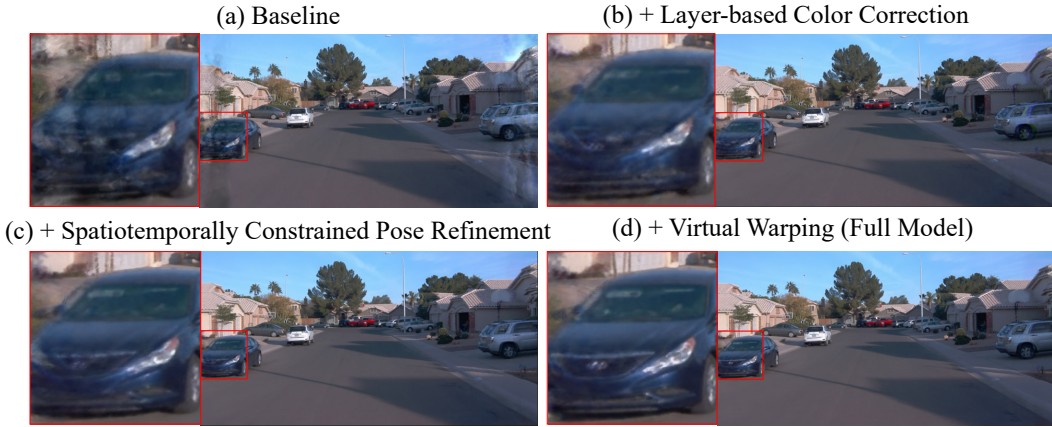

(c) + Spatiotemporally Constrained Pose Refinement     (d) + Virtual Warping (Full Model)

Figure 7: Each proposed module enhances the rendering quality of novel views.

GT            W/O Virtual Warping        W/ Virtual Warping

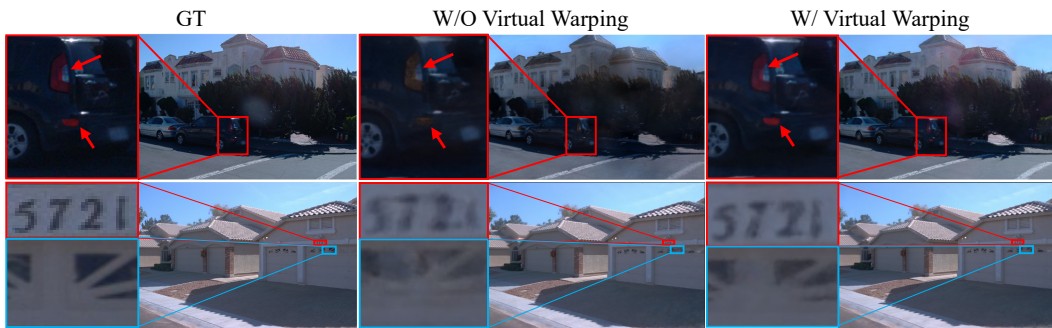

Figure 8: Our virtual warping contributes to color correction (top) and edge sharpness (bottom).

**Efficiency Analysis** We compare the efficiency of different methods in Tab. 3. All methods are tested on one NVIDIA Tesla V100 GPU with an image resolution of $1920 \times 1280$. Note that our training time includes both steps (pose refinement and NeRF training) described in Sec. 3.5. Zip-NeRF is more efficient than other methods except Instant-NGP, which is specifically designed for NeRF acceleration. Since our method is built upon Zip-NeRF, our method consumes a bit more time than Zip-NeRF but achieves a significant improvement in rendering quality.

Table 3: Efficiency Analysis. Tested on one NVIDIA Tesla V100 GPU with image resolution $1920 \times 1280$.

| Method | Training | Inference | PSNR |
|---|---|---|---|
| Mip-NeRF (Barron et al. (2021)) | 20h | 70s | 22.42 |
| Mip-NeRF-360 (Barron et al. (2022)) | 14h | 42s | 24.46 |
| Instant-NGP (Müller et al. (2022)) | 30min | 0.35s | 23.84 |
| S-NeRF (Xie et al. (2023)) | 15h | 80s | 24.89 |
| Zip-NeRF (Barron et al. (2023)) | 2h | 2s | 26.21 |
| UC-NeRF (Ours) | 3h | 3.2s | 28.13 |

## 5   CONCLUSION AND FUTURE WORK

In conclusion, we propose UC-NeRF that effectively addresses the challenges of integrating multi-camera systems into the NeRF paradigm by a layer-based color correction module, a virtual warping module, and a spatiotemporally constrained pose refinement module. Experiments on the Waymo and NuScenes datasets demonstrate a significant improvement in rendering quality, setting a new benchmark for neural rendering within multi-camera setups. With the high-quality novel view rendering of our UC-NeRF, the synthesized images can provide plentiful training data for depth estimation, semantic segmentation, and object detection in autonomous driving scenarios. With the development of more neural reconstruction techniques, such as 3D Gaussian splatting ( Kerbl et al. (2023)), the strategies of layer-based correction, virtual warping, and pose refinement in our UC-NeRF can be easily extended to facilitate the reconstruction quality using these new techniques.

**Acknowledgements** Thanks to DJI for providing sufficient computing resources. This work was supported by National Natural Science Foundation of China (NSFC) under Grants 62076230 and the Fundamental Research Funds for the Central Universities under Grant WK3490000008.

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

# A   APPENDIX

## A.1   IMPLEMENTATION DETAILS

We train our UC-NeRF for 40k iterations using Adam optimizer with a batch size of $32384$. The learning rate is logarithmically reduced from $0.008$ to $0.001$, with a warm-up phase consisting of $5000$ iterations.

**Layer-based Color Correction**   In UC-NeRF, we model a scene as the foreground and sky. The foreground is represented by Zip-NeRF while the sky is modeled by the vanilla NeRF ( Mildenhall et al. (2021)). The weight of sky loss is set to $2 \times 10^{-3}$. The dimension of sky latent code and foreground latent code is set to 4. For the MLP that decodes the latent code, we use three layers with 256 hidden units. The weight of transformation regularization is set to $2 \times 10^{-3}$.

**Virtual Warping**   For virtual warping, we randomly sample 9 virtual poses for each existing pose. The occlusion needs to be considered in the case of multiple pixels of the known view warping to the same pixel in the virtual view, which is shown in the red boxes of Fig. 9. We resolve this conflict by taking the warped pixel with the smallest depth value. For generating the geometric consistency mask, we set 6 target views to check the depth consistency. Only pixels with depth absolute relative error within the range of $0.01$ for at least $4$ neighboring views are retained. For each training batch, we sample the rays from the real and virtual images at a ratio of $4 : 1$ respectively.

**Spatiotemporally Constrained Pose Refinement**   We use the reprojection error to optimize the camera poses. To calculate the reprojection error, the feature points need to be extracted from images. We use Superpoint ( DeTone et al. (2018)) to detect and describe the keypoints. The keypoints are matched by mutual nearest neighbors and the confidence higher than $0.95$ is preserved. For each view, we match it with the subsequent ten frames captured by the same camera and the subsequent twenty frames from different cameras. Image pairs with more than 30 matching points are retained.

## A.2   SPECIAL CASE FOR SPATIOTEMPORAL CONSTRAINT

In Sec. 3.4, we model the pose of $kth$ camera at timestamp $i$ as $\mathbf{T}_k^i = \mathbf{T}^i \Delta \mathbf{T}_k$. Here pose refers to the transformation from the camera coordinate to the world coordinate. $\mathbf{T}^i$ refers to the car's ego pose at time $i$. $\Delta \mathbf{T}_k$ refers to the transformation from the $kth$ camera's coordinate to the ego coordinate, which is temporally consistent. During bundle adjustment, $\mathbf{T}_k^i$, and $\mathbf{T}_l^j$ are optimized as Eq. 7 in Sec. 3.4, where $(\mathbf{T}_l^j)^{-1}\mathbf{T}_k^i$ are expressed as:

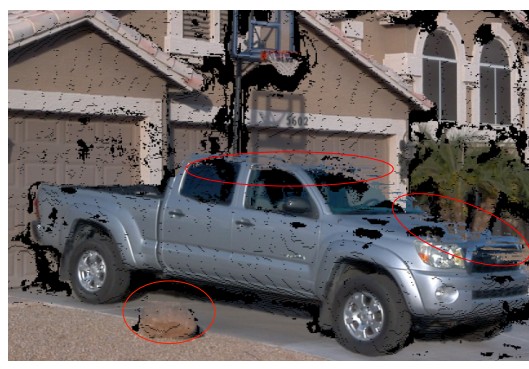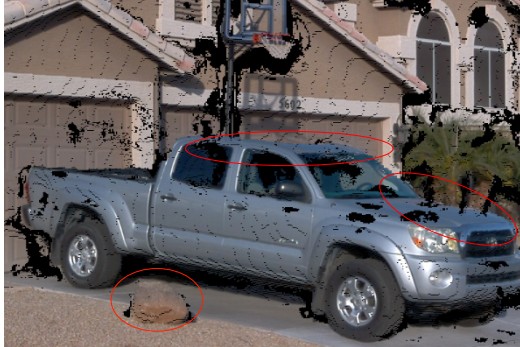

| Vanilla Warping | Occlusion-aware Warping |
|---|---|

Figure 9: Occlusion problem in warping.

$$(\mathbf{T}_l^j)^{-1}\mathbf{T}_k^i = \left(\mathbf{T}^j \Delta \mathbf{T}_l\right)^{-1} \mathbf{T}^i \Delta \mathbf{T}_k$$
$$= \Delta \mathbf{T}_l^{-1} (\mathbf{T}^j)^{-1} \mathbf{T}^i \Delta \mathbf{T}_k \tag{9}$$

Expressing $(\mathbf{T}^j)^{-1}\mathbf{T}^i = \begin{bmatrix} \mathbf{R}^{i,j} & \mathbf{t}^{i,j} \\ \mathbf{0} & \mathbf{1} \end{bmatrix}$, $\Delta \mathbf{T}_l = \begin{bmatrix} \Delta \mathbf{R}_l & \Delta \mathbf{t}_l \\ \mathbf{0} & \mathbf{1} \end{bmatrix}$, and $\Delta \mathbf{T}_k = \begin{bmatrix} \Delta \mathbf{R}_k & \Delta \mathbf{t}_k \\ \mathbf{0} & \mathbf{1} \end{bmatrix}$, then

$$(\mathbf{T}_l^j)^{-1}\mathbf{T}_k^i = \begin{bmatrix} \Delta \mathbf{R}_l^\top & -\Delta \mathbf{R}_l^\top \Delta \mathbf{t}_l \\ \mathbf{0} & \mathbf{1} \end{bmatrix} \begin{bmatrix} \mathbf{R}^{i,j} & \mathbf{t}^{i,j} \\ \mathbf{0} & \mathbf{1} \end{bmatrix} \begin{bmatrix} \Delta \mathbf{R}_k & \Delta \mathbf{t}_k \\ \mathbf{0} & \mathbf{1} \end{bmatrix}$$
$$= \begin{bmatrix} \Delta \mathbf{R}_l^\top \mathbf{R}^{i,j} \Delta \mathbf{R}_k & \Delta \mathbf{R}_l^\top \mathbf{R}^{i,j} \Delta \mathbf{t}_k + \Delta \mathbf{R}_l^\top \mathbf{t}^{i,j} - \Delta \mathbf{R}_l^\top \Delta \mathbf{t}_l \\ \mathbf{0} & \mathbf{1} \end{bmatrix}. \tag{10}$$

When the vehicle is moving straight without any rotation, $\mathbf{R}^{i,j}$ equals to the identity matrix. Thus, the Eq. 10 is simplified as:

$$(\mathbf{T}_l^j)^{-1}\mathbf{T}_k^i = \begin{bmatrix} \Delta \mathbf{R}_l^\top \Delta \mathbf{R}_k & \Delta \mathbf{R}_l^\top \Delta \mathbf{t}_k + \Delta \mathbf{R}_l^\top \mathbf{t}^{i,j} - \Delta \mathbf{R}_l^\top \Delta \mathbf{t}_l \\ \mathbf{0} & \mathbf{1} \end{bmatrix}. \tag{11}$$

If the image correspondences are not established across cameras, $i.e.$, $l = k$, then Eq. 11 can further simplified as:

$$(\mathbf{T}_l^j)^{-1}\mathbf{T}_k^i = \begin{bmatrix} \mathbf{I} & \Delta \mathbf{R}_k^\top \mathbf{t}^{i,j} \\ \mathbf{0} & \mathbf{1} \end{bmatrix}, \tag{12}$$

which suggests that the relative transformation between any two neighboring poses of the same camera $k$ remains unaffected by $\Delta \mathbf{t}_k$, thus resulting in a lack of constraint on the camera's translation $\Delta \mathbf{t}_k$ during the optimization process. This implies that the image correspondences across both cameras and timestamps ensure a robust constraint on inter-camera transformation.

## A.3 Experiments

### A.3.1 Application: Synthesized Views for Monocular Depth Estimation

With the obtained 3D NeRF, we can generate additional photo-realistic images from novel viewpoints. The synthesized images can facilitate downstream perception tasks like monocular depth estimation. We first train VA-DepthNet, a state-of-the-art monocular depth estimation model (Liu et al. (2023)), on the original real images. We then train the model by combining the original real images and the new synthesized images (VA-DepthNet*). As Tab. 4 illustrates, the accuracy of the estimated depth is improved with such a data augmentation. Fig. 10 also shows such an operation leads to sharper edges and more accurate predictions.

Table 4: The accuracy of depth estimation using VA-DepthNet before and after adding our rendered novel views (VA-DepthNet*) for training.

| Method | Abs Rel ↓ | RMSE ↓ | $\delta 1$ ↑ |
|---|---|---|---|
| VA-DepthNet | 0.078 | 2.82 | 93.7 |
| VA-DepthNet* | **0.076** | **2.64** | **94.2** |

| Image | VA-DepthNet | VA-DepthNet* | Image | VA-DepthNet | VA-DepthNet* |
|---|---|---|---|---|---|

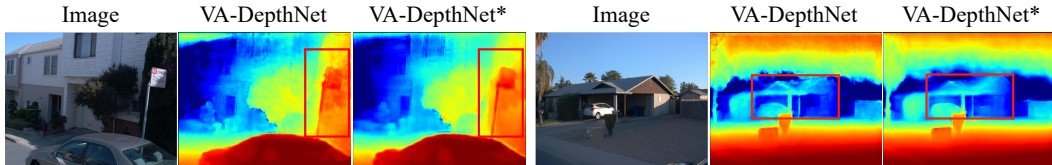

Figure 10: Compared to training VA-DepthNet (Liu et al. (2023)) on the original data, augmenting it with our rendered novel views (VA-DepthNet*) leads to improved depth estimation. Depth of VA-DepthNet* in the red boxes exhibits sharper edges and smoother surfaces.

### A.3.2 MORE ABLATION STUDY RESULTS

**Color Correction Strategies.** We compare our color correction with other strategies that also model the image-dependent appearance with latent codes, as done in NeRF in the wild ( Martin-Brualla et al. (2021)) and Urban-NeRF ( Rematas et al. (2022)). As shown in Fig. 11, the baseline (a) exhibits noticeable color discontinuities in regions where camera views overlap (indicated by red arrows). Although NeRF in the wild (b) models image-dependent appearance through latent codes, the absence of constraints on latent codes results in the disentanglement of attributes not related to the cause of color inconsistency. As a result, the panoramic rendering produced by it exhibits significant blurriness on both sides (emphasized by red boxes), along with additional texture artifacts in the sky (red arrows). Urban-NeRF (c) also decodes the latent code into affine transformations to model image-dependent appearance. However, due to the absence of separate modeling for color transformations across different image regions, color discontinuities, as indicated by the red arrows, persist in the overlapping regions of the cameras. Additionally, the areas observed by different cameras are not unified into the same color space. When employing a single latent code for rendering the panorama image, the color within the central region adheres closely to reality. However, severe color deviations occur in the peripheral area, such as the grass in the red box, which appears black instead of its natural color. In contrast, our method addresses the color inconsistencies and ensures clear details with consistent colors. Tab. 5 also demonstrates that our approach achieves the best rendering results.

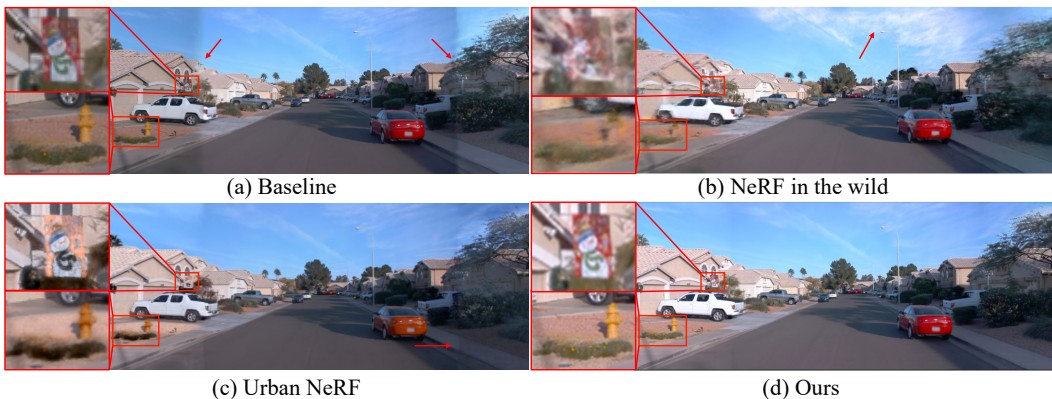

Figure 11: Comparison of color correction strategies for rendering images with large field of view.

Table 5: Comparison of different strategies for color correction.

| Method | PSNR ↑ | SSIM ↑ | LPIPS ↓ |
|---|---|---|---|
| NeRF in the wild ( Martin-Brualla et al. (2021)) | 25.59 | 0.839 | 0.389 |
| Urban-NeRF ( Rematas et al. (2022)) | 27.89 | 0.849 | 0.378 |
| Ours | **28.15** | **0.851** | **0.374** |

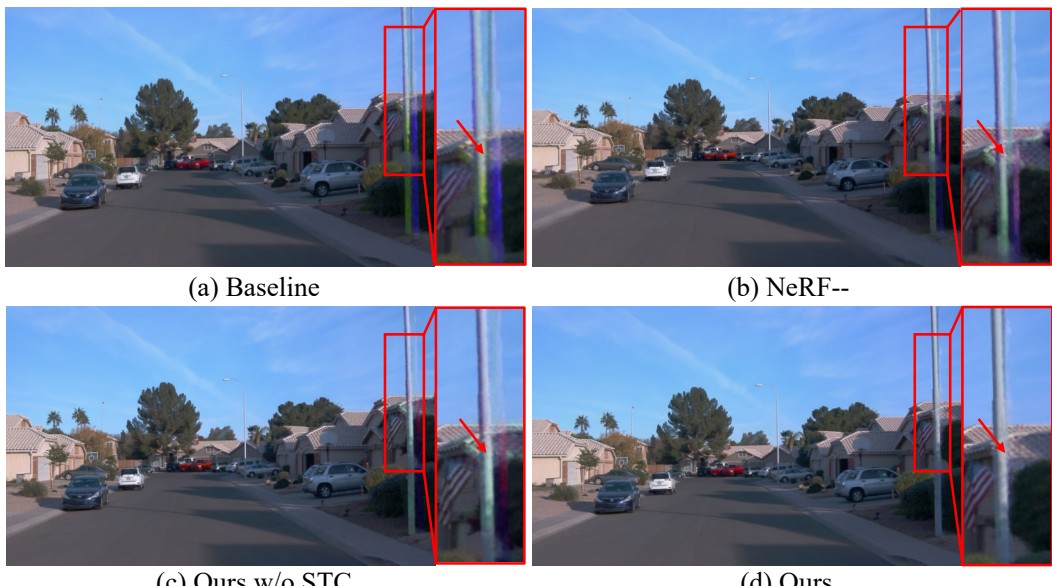

(a) Baseline            (b) NeRF--

(c) Ours w/o STC          (d) Ours

Figure 12: Quantitative comparison of different pose optimization strategies. We effectively eliminate the ghosting of the pole. STC refers to the proposed spatiotemporal constraint.

**Pose Refinement Strategies** S-NeRF ( Xie et al. (2023)) explores the performance of various NeRF algorithms that jointly optimize poses in urban scenes and found that NeRF−− ( Wang et al. (2021)) yields the best results. Thus, we compare our method with NeRF−− and validate the significance of the spatiotemporal constraint proposed in our paper. As demonstrated in Tab. 6, NeRF−− indeed enhances rendering results compared to the baseline which does not refine poses, and performance does not exhibit a significant change with the inclusion of the spatiotemporal constraint. In contrast, our spatiotemporally constrained pose refinement achieves a remarkable 238% improvement compared to NeRF−−. Fig. 12 clearly demonstrates how our method effectively resolves rendering artifacts. Due to color variation among different cameras, NeRF−− struggles to achieve precise pose optimization based on the photometric error. The ghosting of the pole (b) does not change significantly compared to the baseline (a). However, when poses are optimized by explicit pixel correspondences, the ghosting is noticeably reduced (c). Furthermore, with the addition of our spatiotemporal constraint, the artifact completely disappears (d).

Table 6: Ablation study on different strategies for pose refinement. NeRF−− refine poses within the NeRF framework by photometric loss while we refine poses based on explicit pixel correspondences among images. STC refers to the proposed spatiotemporal constraint.

| Method | PSNR ↑ | SSIM ↑ | LPIPS ↓ |
|---|---|---|---|
| Baseline | 28.09 | 0.851 | 0.374 |
| NeRF−− | 28.48 | 0.851 | 0.383 |
| NeRF−− w/ STC | 28.51 | 0.852 | 0.383 |
| Ours w/o STC | 29.01 | 0.866 | 0.376 |
| Ours | **29.14** | **0.867** | **0.355** |

Table 7: Ablation study on the weight of sky loss.

| $w_{sky}$ | PSNR ↑ | SSIM ↑ | LPIPS ↓ |
|---|---|---|---|
| 0 | 27.89 | 0.849 | 0.378 |
| 0.001 | 28.05 | 0.851 | 0.379 |
| 0.002 | **28.15** | **0.851** | **0.374** |
| 0.004 | 27.98 | 0.850 | 0.378 |

Table 8: Comparison of diverse weather conditions. (Waymo Segment-100170, Waymo Segment-150908)

| Method | Rainy | | | Night | | |
|---|---|---|---|---|---|---|
| | PSNR ↑ | SSIM ↑ | LPIPS ↓ | PSNR ↑ | SSIM ↑ | LPIPS ↓ |
| Zip-NeRF[3] | 27.65 | 0.831 | 0.434 | 30.69 | 0.864 | 0.512 |
| UC-NeRF (Ours) | **30.03** | **0.866** | **0.387** | **31.32** | **0.869** | **0.491** |

**Weight for Sky Loss**    As shown in Tab. 7, we compare the performance of our UC-NeRF using different weights for sky loss. Our UC-NeRF is not very sensitive to the changes in the loss weights. Using a large weight of sky loss might diminish the weight of the photometric loss, leading to a slight performance decline. $[0.001, 0.002]$ is the reasonable range for our loss weight.

**Results on diverse weather conditions**    We further validate the robustness of our method under night and rainy conditions. As shown in Tab. 8, our approach still significantly outperforms

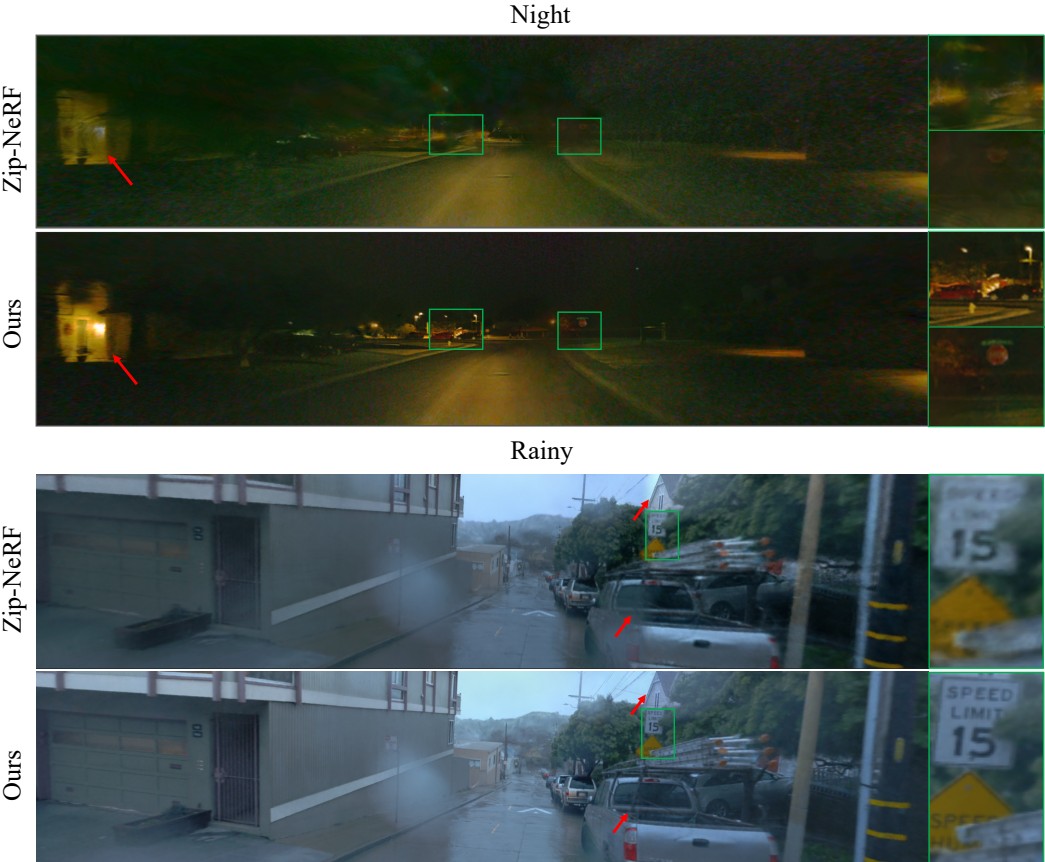

Figure 13: 180° paranoma rendering results in night and rainy conditions.

ZipNeRF in these scenarios. As illustrated in Fig. 13, we manage to eliminate the rendering artifacts caused by color inconsistency (indicated by red arrows) and achieve better rendering of details (highlighted by the green boxes).

**More Results on Layer-based Color Correction**   As shown in Fig 14, we present additional rendering results in overlap regions of different cameras. The original NeRF exhibits significant color inconsistencies in overlapping areas. Through color correction, we are able to render images that maintain global color consistency.

**More Results on Spatiotemporally Constrained Pose Refinement**   As shown in Fig 15, the issue of rendering artifacts and blurriness caused by pose errors is quite common in the multi-camera setup. Based on our observations, they typically occur in the overlapping regions captured by different cameras. This implies that these rendering artifacts result from errors in the relative transformations between the cameras. By explicitly modeling the relative transformations between cameras and ensuring spatiotemporal constraints during optimization, it is evident that these problems have been significantly addressed (shown in red boxes).

| w/o Color Correction | w/ Color Correction |

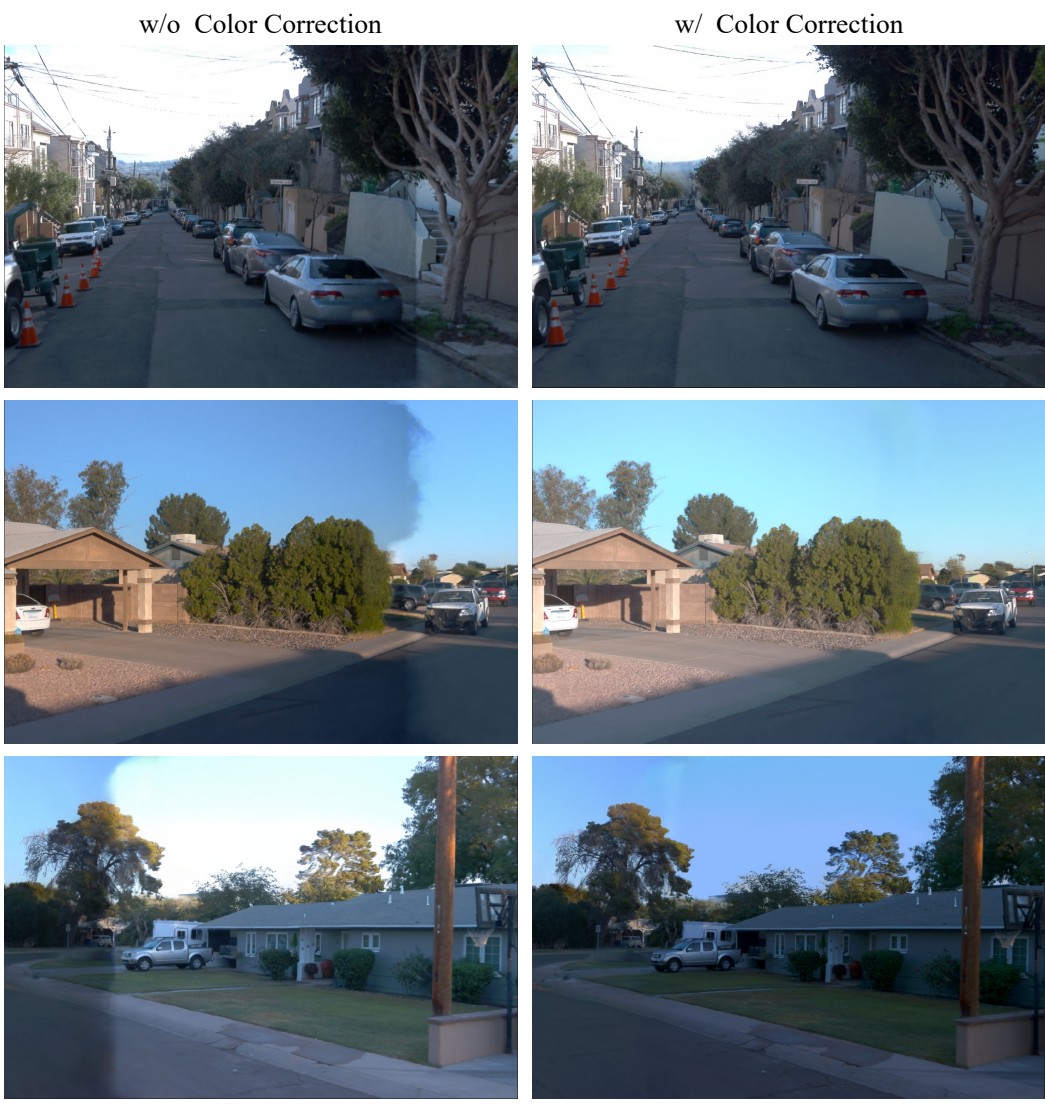

Figure 14: More results on color correction.

w/o  Pose Refinement                                      w/ Pose Refinement

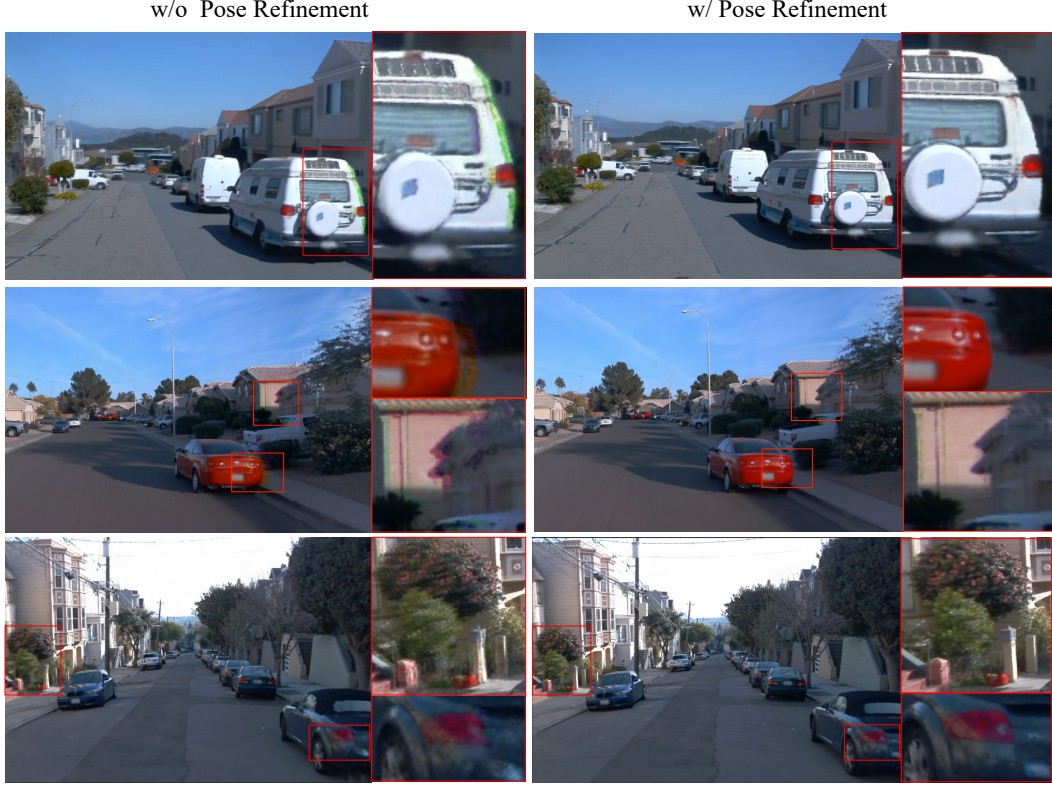

Figure 15: More results on pose refinement.

### A.3.3  MORE RESULTS ON WAYMO AND NUSCENES

We further present rendering results on the Waymo and NuScenes datasets, which are compared with state-of-the-art methods S-NeRF ( Xie et al. (2023)) and Zip-NeRF ( Barron et al. (2023)). As shown in Fig. 16- 18, it is evident that both S-NeRF and Zip-NeRF exhibit color inconsistencies on the sides of the images, where they overlap with other cameras. In contrast, we have addressed this issue through layer-based color correction. Furthermore, we achieve improved rendering quality, such as clear contours, patterns, and text, and more accurate geometry by incorporating virtual warping and spatiotemporally constrained pose refinement.

S-NeRF   Zip-NeRF   Ours

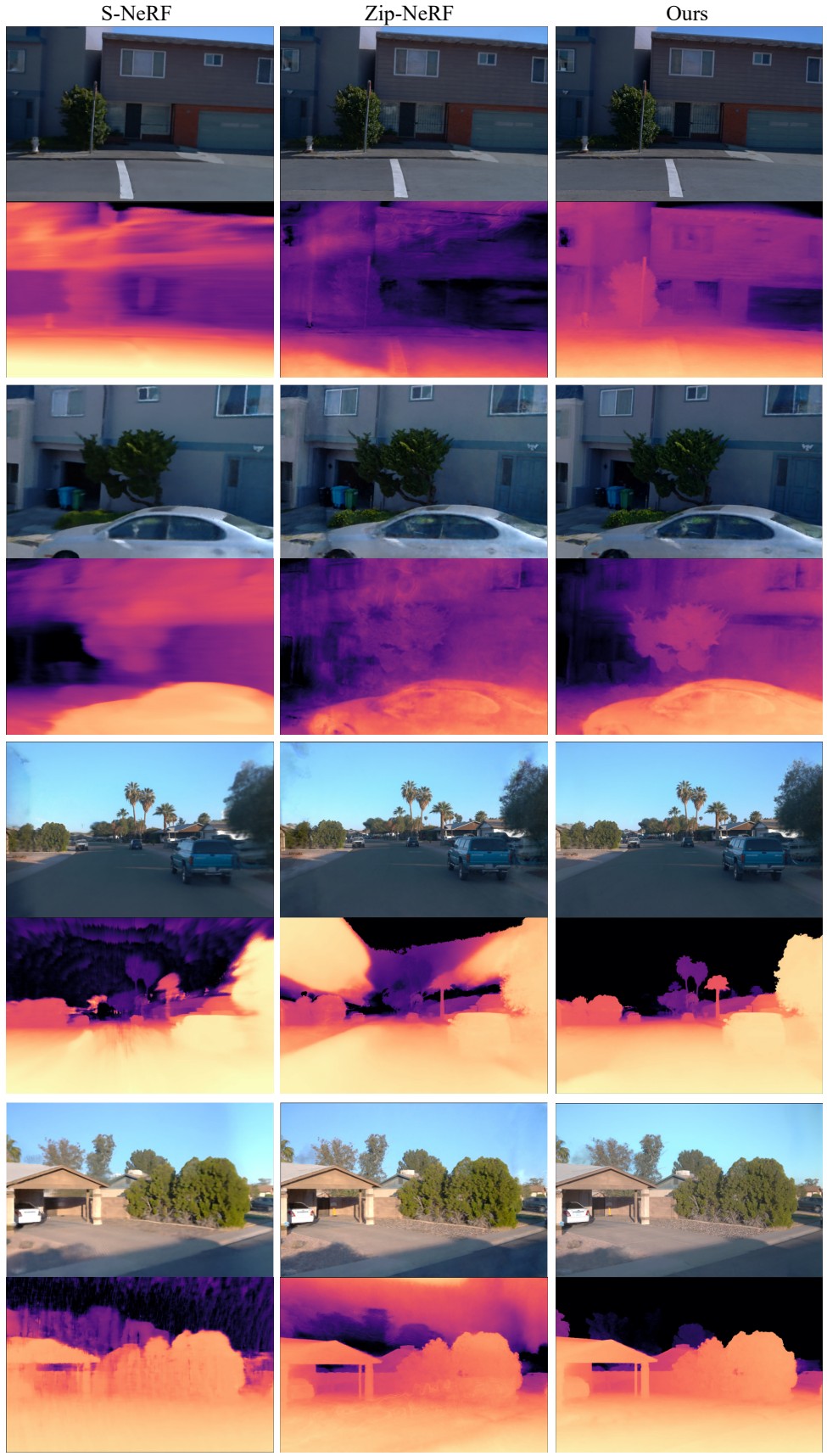

Figure 16: Comparison of the rendering results with the state-of-the-art S-NeRF and Zip-NeRF in Waymo.

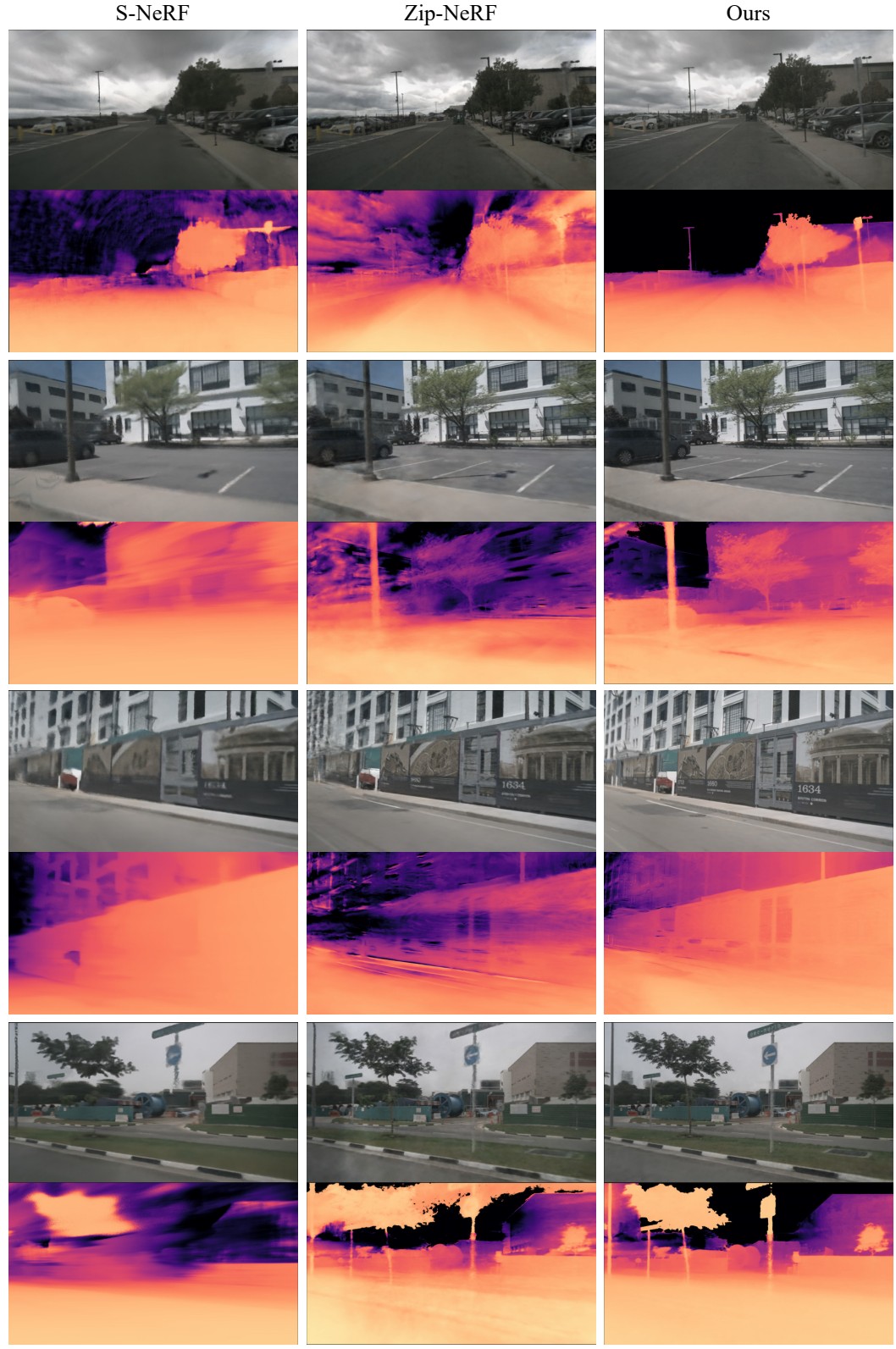

Figure 17: Comparison of the rendering results and dept maps with S-NeRF Tosi et al. (2023) (left) and Zip-NeRF Barron et al. (2023) (middle) in NuScenes.

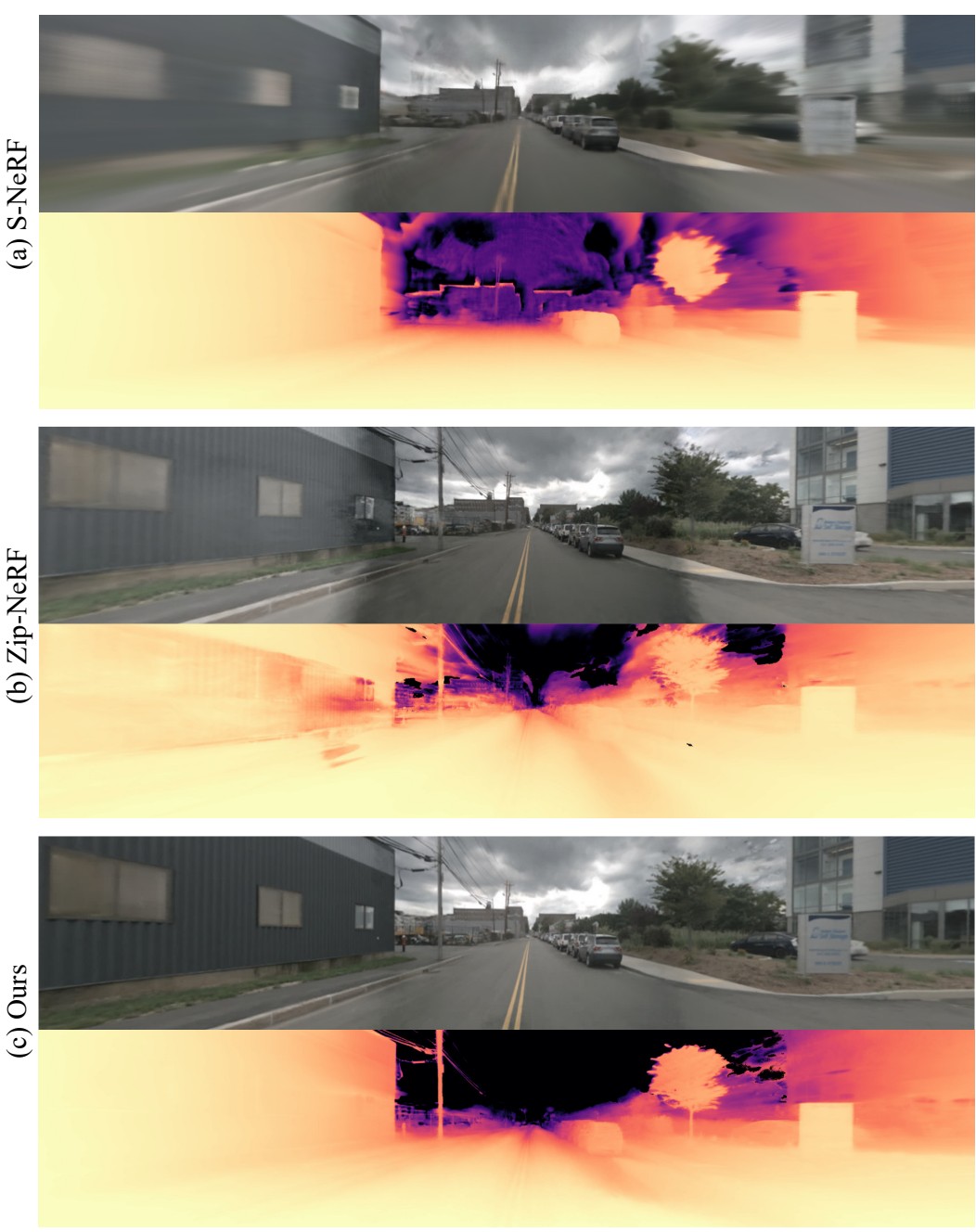

Figure 18: 180° paranoma rendering results in NuScenes.

