# OpenReview forum: "UC-NERF: Neural Radiance Field for Under-Calibrated Multi-View Cameras in Autonomous Driving"
_ICLR.cc/2024/Conference — ICLR 2024 poster_

### Official Review · Reviewer_ehJ8 · 2023-10-31

**Soundness:** 2 fair
**Presentation:** 3 good
**Contribution:** 2 fair
**Rating:** 6
**Confidence:** 5

**Summary:**

This work proposed UC-NeRF, a method for novel view synthesis in under-calibrated multi-view camera systems. The authors propose a three-step approach to address these challenges:

1. Layer-based color correction: This step rectifies the color inconsistency in different image regions by applying color correction to each layer of the image pyramid.

2. Virtual warping: This step generates more viewpoint-diverse but color-consistent virtual views for color correction and 3D recovery. The authors show that virtual warping benefits color correction and edge sharpness.

3. Spatiotemporally constrained pose refinement: This step is designed for more robust and accurate pose calibration in multi-camera systems. The authors demonstrate that this step improves the accuracy of depth estimation in large-scale outdoor scenes.

The paper includes experimental results on several datasets and comparisons with other methods. The authors show that UC-NeRF achieves state-of-the-art performance in novel view synthesis and improves the sensing capabilities of multi-camera systems.

**Strengths:**

- The problem setting is interesting. The proposed method can be an enhancement of current NeRF techniques.

- The proposed method is sound and effective, regardless of its simplicity - complexity is not a criterion for us to judge whether a paper is good or not.

- The authors did exhaustive experiments to show the effectiveness of the proposed method. Ablation studies also show the effectiveness of each module.

**Weaknesses:**

- I think the introduction of this paper is not well written. It took me some time to understand why this work needs virtual warping and what are the differences between single-camera NeRF and multi-camera NeRF (since each camera in a multi-camera system can be deemed as a single camera).

- The virtual warping step relies on the MVS method to generate dense depth maps, which may not generalize to street views (I'm not certain about this) and may need further pertaining.

- In Eq.(5), the author does not explain what is $\mathbf{b}$ and $\mathbf{d}$ denote.

- In Eq. (7), it is unclear whether the relative transformation $\Delta \mathbf{T}_k$ is optimized.

- The final training loss is missing, e.g. $\mathcal{L} = \mathcal{L}_{\text{pho}} + \lambda_1 \mathcal{L}_{\text{reg}} +  \lambda_1 \lambda_2 \mathcal{L}_{\text{rpj}} $.

- The pose refinement step is quite straightforward. Since the relative pose constraints $\Delta \mathbf{T}$ in the same rig can be obtained through calibration, I think it is naive to decompose the camera pose into the ego pose and a relative transformation $\Delta \mathbf{T}$. Moreover, the pose refinement step requires point correspondences, which could introduce outliers since it is well known that SOTA point matching methods are prone to repetitive structures and moving objects.

**Questions:**

- The pose refinement step relies on keypoints, which could be a shortcoming. Did the author consider DBARF (CVPR 2023) and FlowCam (NeurIPS 2023), which jointly optimize consecutive camera poses and NeRF?
Actually, due to the vibrations during driving, the relative camera poses in a rig could change. I think the author mentioned it in the introduction, but the case is not handled in the formulation (Eq. (7)).

- Did you reimplement Zip-NeRF or use others' reimplementation of Zip-NeRF? If it is the latter case, the URL should be provided since Zip-NeRF does not release its code.

---

> ### Author Response · Authors · 2023-11-19
> **Response to Reviewer ehJ8 (Part 1/2)**
>
> **1) The writing of introduction.**
>
> Thanks for your suggestions. We have revised the introduction to better understand the multi-camera system and the motivation for virtual warping.
>
> **2) The usage of MVS methods.**
>
> We use the public model of CER-MVS without finetuning. Virtual warping can enhance color correction and rendering quality, regardless of the depth sparsity. We also demonstrate its effectiveness with sparse lidar input in Tab. 1 below.
>
> Table 1. The performance of rendering with different sources of depth for virtual warping. Tested on a randomly selected Waymo scene.
> | Depth source| PSNR ↑   | SSIM ↑   | LPIPS ↓ |
> |-------|-------|-------|-------|
> |No virtual warping | 28.74| 0.874 | 0.338|
> |Lidar virtual warping | $\underline{29.04}$ | $\underline{0.879}$ | $\underline{0.335}$ |
> | MVS virtual warping (Ours) | $\mathbf{29.49}$ | $\mathbf{0.881}$ | $\mathbf{0.334}$|
>
> **3) The explanation of $\textbf{b}$ and $\textbf{d}$ in Eq.5.**
>
> Thanks for your suggestion. In Eq. 5, $\textbf{b}$ and $\textbf{d}$ should be $\textbf{x}$ and $\textbf{y}$ respectively, which represent the translation term of the color correction matrix. We have revised them in the paper.
>
> **4) Whether the relative transformation is optimized?**
>
> Yes. The relative transformation is optimized.
>
> **5) The final training loss is missing.**
>
> Thanks for your suggestion. We add the final training loss in Sec. 3.5 of the revised paper.
>
> **6) The pose refinement step relies on key points, which could introduce outliers.**
>
> We have included the RANSAC[1] algorithm for filtering outliers before bundle adjustment.
> Generally, pose optimization is based on point correspondences or photometric consistency. However, images taken by different cameras have significant color differences, which violates the assumption of photometric consistency. Furthermore, joint optimization of NeRF, color correction, and poses based on photometric loss only might not provide sufficient constraints.
>
> In contrast, urban scenes are typically endowed with rich textures, which allow for the extraction of a substantial number of key points. We posit that the establishment of image correspondences serves as a more effective and direct strategy for pose optimization. We demonstrate that optimizing poses through explicit pixel correspondences is better than NeRF-pose joint optimization with photometric loss in the multi-camera setting in Tab. 6 and Fig. 12 of our revised paper (in Appendix A.3.2).
>
> **7) The consideration of DBARF[2] and FlowCam[3].**
>
> Thanks for your suggestion. We finetune FlowCam and DBARF to our Waymo scenes, with results in Tab. 2 below. The performance of our method is better than FlowCam and DBARF in a multi-camera setting. DBARF optimizes poses through cost feature maps between images, which can be affected by significant color differences in images of different cameras. FlowCam requires the calculation of optical flow for pose optimization. However, given the restrained overlap between images from different cameras, the effectiveness of current optical flow techniques and subsequent pose estimation is limited in a multi-camera setting.
>
> Table 2. Comparisons of our method with FlowCam and DBARF. Tested on a randomly selected Waymo scene.
> | Method | PSNR ↑   | SSIM ↑   | LPIPS ↓ |
> |-------|-------|-------|-------|
> |DBARF | 23.03 | 0.672 | 0.472 |
> |FlowCam | 23.50 | 0.721 | 0.401 |
> |UC-NeRF (Pose Refinement Only) | $\textbf{27.79}$ | $\textbf{0.867}$ | $\textbf{0.363}$|

---

> > ### Comment · Reviewer_ehJ8 · 2023-11-21
> > **Thanks for the reply**
> >
> > Thanks for the detailed reply from the authors. Your reply answered most of my questions. However, I'm still worried that the point correspondence could be a limitation. And some of your explanations may not be accurate:
> > - "We have included the RANSAC[1] algorithm during bundle adjustment". I think the RANSAC is used to filter wrong correspondences before the bundle adjustment instead of during the bundle adjustment since traditional RANSAC is not differentiable.
> > - "which is robust to the outliers of key points." I think in dynamic scenes, RANSAC is not enough to filter those wrong matches, e.g. matches on a moving vehicle or pedestrians. Since your experiments are conducted on urban street views, I think it could be a problem.
> > Anyway, this is just my concern, and I think it could be improved in the future. Therefore, I will keep my score unchanged.

---

> ### Author Response · Authors · 2023-11-19
> **Response to Reviewer ehJ8 (Part 2/2)**
>
> **8) The change of the relative camera poses.**
>
> Thanks for your suggestion. As you pointed out, the relative pose between cameras may change during driving, but such considerable changes are typically caused by accumulated shakes over a vehicle's long-term drive like months. NeRF models generally operate on short video sequences lasting from minutes to hours, during which the changes in the relative poses between cameras are subtle.
> On the contrary, directly modeling the transformation between cameras without our proposed spatiotemporal constraint is feasible, but it may lead to less accurate pose estimation. This is primarily because of the fact that images captured by a moving vehicle share limited overlaps, especially among the images captured by different cameras. This could potentially result in insufficient connections between images taken by different cameras, making the pose optimization relatively less constrained and unstable. Considering the two factors, we propose to utilize the spatiotemporal constraint to enhance the pose refinement.
>
> We add an experiment in Tab. 3 below to analyze the spatiotemporal constraint. The experiment is conducted in ten commonly used self-driving sequences without any special selection. Compared to directly modeling the transformation without any constraint, our spatiotemporal constrained pose refinement leads to higher rendering quality of the reconstructed NeRF with more accurate camera poses. Fig. 12 of our revised paper also illustrates the improvement of rendering from this constraint.
>
> Table 3. Effectiveness of spatiotemporal constraint tested on ten Waymo scenes.
> | Method  | PSNR ↑   | SSIM ↑   | LPIPS ↓ |
> |-------|-------|-------|-------|
> | W/O Spatiotemporal Constraint | 27.89 | 0.835 | 0.368|
> | W Spatiotemporal Constraint  | **28.13** | **0.842** |**0.356**|
>
> **9)  Reference on ZipNerf.**
>
> Thanks. We have cited the reimplementation of Zip-NeRF that we use in the revised paper.
>
> ---
>
> [1] Fischler M A, Bolles R C. Random sample consensus: a paradigm for model fitting with applications to image analysis and automated cartography[J]. Communications of the ACM, 1981, 24(6): 381-395.
>
> [2] Chen Y, Lee G H. DBARF: Deep Bundle-Adjusting Generalizable Neural Radiance Fields[C]//Proceedings of the IEEE/CVF Conference on Computer Vision and Pattern Recognition. 2023: 24-34.
>
> [3] Smith C, Du Y, Tewari A, et al. FlowCam: Training Generalizable 3D Radiance Fields without Camera Poses via Pixel-Aligned Scene Flow[J]. Advances in Neural Information Processing Systems, 2023

---

> ### Author Response · Authors · 2023-11-21
> **Thanks for your valuable feedback**
>
> Thank you very much for your valuable feedback. Here is our reply.
>
> * Thank you for your correction. We have revised our response to "before".
>
> *  We appreciate your mention of the challenge posed by dynamic objects. Pose estimation under dynamic scenes is a specialized task, where many solutions are based on key points. For example, Mask-SLAM[1] and DynaSLAM[2] filter out key points in areas that may be dynamic objects (vehicles and pedestrians) through semantic segmentation. As our paper mainly addresses the challenges brought by introducing multiple cameras, we did not initially consider the issue of dynamic objects. But the solutions for filter dynamic objects, such as the semantic segmentation, can be easily added to our pose optimization module.
>
> Finally, we truly appreciate your recognition and suggestions for our work.
>
> ---
>
> [1] Kaneko M, Iwami K, Ogawa T, et al. Mask-SLAM: Robust feature-based monocular SLAM by masking using semantic segmentation[C]//Proceedings of the IEEE conference on computer vision and pattern recognition workshops. 2018: 258-266.
>
> [2]Bescos B, Fácil J M, Civera J, et al. DynaSLAM: Tracking, mapping, and inpainting in dynamic scenes[J]. IEEE Robotics and Automation Letters, 2018, 3(4): 4076-4083.

---

### Official Review · Reviewer_oEhA · 2023-11-01

**Soundness:** 3 good
**Presentation:** 3 good
**Contribution:** 4 excellent
**Rating:** 6
**Confidence:** 4

**Summary:**

The paper presents UC-NeRF, a method for new view image synthesis in multicamera systems. They introduce models for color correction, virtual warping, and pose refinement to improve upon the results of Zip-NeRF and NeRF. Each of these operations defines a loss function L_{sky}, L_{reg}, and L_{rpj}. The results seem to suggest that they are achieving state-of-the-art results. Yet, the code to verify this claim is not available. Further details may be needed to implement their ideas completely.

**Strengths:**

The paper presents state-of-the-art results. They benchmark the performance of UC-NeRF with several other strategies that have been recently introduced. Their ablation study suggests that each term in the loss function improves the results.

**Weaknesses:**

It would be great if the authors could share their code; even promising to share upon acceptance will be understandable. NeRF code is readily available. Otherwise, the authors should increase the clarity of their presentation to explain how their ideas could be implemented and the results reproduced for verification.

**Questions:**

In 3.5, I understand that UC-NeRF is NeRF trained on the original NeRF’s photometric loss and L_{sky} and L_{reg}, but you are also using L_{rpj}, correct? Is the total loss the sum of the individual losses? Are there weights on the losses before adding them?

What is mathbf{b} and mathbf{d} in (5)?

Define d_v and d_o in (6)

**Details Of Ethics Concerns:**

no ethics concerns

---

> ### Author Response · Authors · 2023-11-19
> **Response to Reviewer oEhA**
>
> **1) The share of our code.**
>
> We promise to share our code upon acceptance.
>
> **2) The formulation of the loss.**
>
> $L_{rpj}$  is used for pose refinement only. The total loss for NeRF training is the weighted sum of the photometric loss, $L_{sky}$ and $L_{reg}$. We add the formulation of the loss and revise the training strategy in Sec. 3.5. Please check out general response 1 about training strategy and loss formulation.
>
> **3) The definition of $\mathbf{b}$, $\mathbf{d}$, $d_v$, $d_o$.**
>
> Many thanks. In Eq. 5, $\mathbf{b}$ and $\mathbf{d}$ should be $\mathbf{x}$ and $\mathbf{y}$ respectively, which represent the translation term of the color correction matrix. In Eq. 6, $d_v$ and $d_o$ refer to the pixel depth in
> virtual views and the corresponding pixel depth in original real views. All required definitions have been added to the paper.

---

> > ### Comment · Reviewer_oEhA · 2023-11-22
> >
> > Thanks for your pledge to share the code. Your commitment is an important point as it increases transparency, gives others the chance to confirm the results, and provides a solid stepping stone for further advancing the field.

---

> ### Author Response · Authors · 2023-11-22
> **Thanks for your reply**
>
> Thanks for your reply and understanding. We will definitely share the code.

---

### Official Review · Reviewer_5Tdv · 2023-11-01

**Soundness:** 3 good
**Presentation:** 3 good
**Contribution:** 2 fair
**Rating:** 5
**Confidence:** 4

**Summary:**

The paper introduces UC-NeRF, a novel approach designed specifically for under-calibrated multi-view camera systems, addressing the challenges faced when applying NeRF techniques in such setups. The method incorporates layer-based color correction, virtual warping, and spatiotemporally constrained pose refinement to achieve exceptional performance in novel view synthesis and enhance the sensing capabilities of multi-camera systems. The contributions of the paper encompass the introduction of a new dataset tailored for under-calibrated multi-view camera systems, a novel layer-based color correction method, and an algorithm for spatiotemporally constrained pose refinement. The effectiveness of UC-NeRF is demonstrated through experiments conducted on the new dataset, and comparisons are made against state-of-the-art methods.

**Strengths:**

S1. This paper is well-written and easy to follow.
S2. The proposed method is technically sound.
S3. The experiment design especially the ablation study is solid and the results are noticeable.

**Weaknesses:**

W1. The novelty of this paper is somewhat limited to me:
W1-1. In terms of the first key innovation, namely layer-based color correction, why we can not use some classical multiple views color correction solutions in the structure-from-motion field as a pre-processing step instead of a module inside the NeRF? It should be justified. Besides, some existing NeRFs also addresses similar problem such as RAWNeRF and block-NeRF, what are the main differences between the proposed method and these works?
W1-2 In terms of the spatiotemporally constrained pose refinement, there are some similar NeRFs that also consider the spatial and temporal connections between cameras for pose optimization. Name a few but not completed lists such as BARF (Lin et al. ICCV 2021) and BAD-Nerf (Wang et al. CVPR 2023). What is the novelty of the proposed method over these works?

W2. The experiment comparisons are limited since only Mip-NeRF was used. Why not compare to some large-scale NeRDs such as block-NeRF or multi-views NeRFs such as MC-NeRF and NeRF-MS. The authors should justify the reason.

**Questions:**

Please check the weaknesses listed above.

---

> ### Author Response · Authors · 2023-11-19
> **Response to Reviewer 5Tdv （Part 1/2)**
>
> **1) Why we can not use some classical multiple views color correction solutions in the structure-from-motion field as a pre-processing step instead of a module inside the NeRF?**
>
> First, limited overlap in images from different cameras hampers classical color correction methods, still causing color inconsistencies in NeRF training.
> Secondly, traditional multi-view color correction methods [1,2] propagate the reference colors to the corresponding 2D pixels in the target frames, which can be challenging to ensure global color consistency across a large set of images. However, NeRF can directly learn the color correction in the 3D space, thus naturally ensuring global color consistency in the rendered 2D images.
>
> **2) What are the main differences between the proposed color correction method and existing works, such as Block-NeRF and Raw-NeRF[3]?**
>
> Existing works, including Block-NeRF, alleviate the inconsistent color supervision by modeling image-dependent appearance with a global latent code for each image, which does not perform well in our setting (as illustrated in Fig. 11 and Tab. 5 of our paper). Compared to them, **(i)** we propose the concept of layer-wise color correction, which approximates color correction in foreground and sky regions as different affine transformations. This not only avoids the disentanglement of attributes unrelated to color inconsistency when the latent code implicitly models the image-dependent appearance, but also ensures the ability to model different color transformations in different regions of a single image.
> **(ii)** we provide a novel module ``Virtual Warping'' to further improve color correction. Learning one color correction for each image leads to overfitting when the training images lack color and viewpoint diversity. Virtual warping provides diverse color observations from novel viewpoints of different cameras for each 3D region, alleviating the overfitting of color correction and improving the 3D recovery.
>
> Raw-NeRF inputs the noisy mosaicked linear raw images from the sensor rather than the common images processed by a camera pipeline. Thus, it cannot be directly applied to our scenario.
>
> **3) What is the novelty of the proposed method over existing NeRF-pose joint optimization methods, such as BARF, and BAD-NeRF[4]?**
>
> Existing methods, including BARF and BAD-NeRF, jointly optimize poses with NeRF by photometric loss. However, they do not consider two difficulties of pose optimization in multi-camera systems.
>
> **(i)** Images taken by multiple cameras exhibit noticeable color inconsistencies. Joint optimization of NeRF, color correction, and poses based on photometric loss only might not provide sufficient constraints. Therefore, we optimize the poses more directly by establishing pixel correspondences. We add an experiment to demonstrate our superiority to BARF in Tab. 1 below. BAD-NeRF is tailored for the rendering of blurring images, which is not comparable with our method.
>
> Table 1. Comparison of our pose refinement method with BARF. Tested on a randomly selected Waymo scene.
> | Method  | PSNR ↑   | SSIM ↑   | LPIPS ↓ |
> |-------|-------|-------|-------|
> |BARF|  28.65 | 0.856 |  0.373 |
> |UC-NeRF (Ours) | **29.14** | **0.867** | **0.355** |
>
> **(ii)** Inaccurate relative transformations between different cameras will cause rendering artifacts. However, directly optimizing the camera transformations without any constraint is unstable and inaccurate in our setting, since the images captured by multiple cameras share limited overlaps. We introduce a constraint that explicitly models the relative transformations between cameras as temporally consistent but optimizable variables to achieve more accurate pose refinement.

---

> ### Author Response · Authors · 2023-11-19
> **Response to Reviewer 5Tdv （Part 2/2)**
>
> **4) The experiment comparisons are limited since only Mip-NeRF was used. Why not compare to some large-scale NeRFs such as block-NeRF or multi-views NeRFs such as MC-NeRF[5] and NeRF-MS[6].**
>
> Thanks for your suggestion. We have compared the strategies used for color correction and pose refinement in Block-NeRF in Tab. 5, Tab. 6, Fig. 11, and Fig. 12 of our revised paper (in Appendix A.3.2).
> Block-NeRF does not release its code. According to its paper, it is built on Mip-NeRF. For a fair comparison, we replace our color correction strategy and pose refinement strategy in ZipNeRF with the strategies used in Block-NeRF. (Block-NeRF uses the color correction strategy from NeRF-in-the-wild and pose refinement strategy same as NeRF$--$.)
>
> We add an experiment in Tab. 2 below, demonstrating our method's superiority over NeRF-MS. NeRF-MS lacks specific modeling of color correction for multi-camera systems, potentially correcting appearance unrelated to color inconsistency and resulting in inferior performance.  MC-NeRF is a contemporaneous work posted on arXiv in September 2023 without public codes. Therefore, we cannot compare it with ours within the limited discussion time. It focuses on optimizing the intrinsics of different cameras during pose optimization while we propose the spatiotemporal constraint between different cameras to enhance pose optimization. The discussion of these works is also updated in the related work of our revised paper.
>
> Table 2. Comparison of our method with NeRF-MS in a randomly selected Waymo scene.
> | Method  | PSNR ↑   | SSIM ↑   | LPIPS ↓ |
> |-------|-------|-------|-------|
> | NeRF-MS | 26.87 | 0.828 | 0.392 |
> | UC-NeRF (Color Correction Only) | **28.15** | **0.851** | **0.374** |
>
> ---
> [1] Ye S, Lu S P, Munteanu A. Color correction for large-baseline multiview video[J]. Signal Processing: Image Communication, 2017, 53: 40-50.
>
> [2] Lu S P, Ceulemans B, Munteanu A, et al. Spatio-temporally consistent color and structure optimization for multiview video color correction[J]. IEEE Transactions on Multimedia, 2015, 17(5): 577-590.
>
> [3] Mildenhall B, Hedman P, Martin-Brualla R, et al. Nerf in the dark: High dynamic range view synthesis from noisy raw images[C]//Proceedings of the IEEE/CVF Conference on Computer Vision and Pattern Recognition. 2022: 16190-16199.
>
> [4] Wang P, Zhao L, Ma R, et al. BAD-NeRF: Bundle Adjusted Deblur Neural Radiance Fields[C]//Proceedings of the IEEE/CVF Conference on Computer Vision and Pattern Recognition. 2023: 4170-4179.
>
> [5] Gao Y, Su L, Liang H, et al. MC-NeRF: Muti-Camera Neural Radiance Fields for Muti-Camera Image Acquisition Systems[J]. arXiv preprint arXiv:2309.07846, 2023.
>
> [6] Li P, Wang S, Yang C, et al. NeRF-MS: Neural Radiance Fields with Multi-Sequence[C]//Proceedings of the IEEE/CVF International Conference on Computer Vision. 2023: 18591-18600.

---

### Official Review · Reviewer_9RAi · 2023-11-04

**Soundness:** 3 good
**Presentation:** 3 good
**Contribution:** 3 good
**Rating:** 6
**Confidence:** 5

**Summary:**

In this work, a novel method tailored for novel view synthesis is proposed for under-calibrated multi-view camera systems. In particular, a layer-based color correction is designed to rectify the color inconsistency in different image regions. To generate more viewpoint-diverse but color-consistent virtual views for color correction and 3D recovery, the authors further propose the virtual warping technique. And a spatiotemporally constrained pose optimization strategy is presented to explicitly model the spatial and temporal connections between cameras for pose optimization. Experiments on the Waymoand and NuScenes datasets show that this work achieves high-quality renderings with a multi-camera system and outperforms other baselines by a large margin.

**Strengths:**

+ The proposed layer-based color correction well addresses color inconsistencies in the training images, especially for those taken by different cameras.
+ The virtual warping strategy naturally expands the range of the training views for NeRF, enhancing its effectiveness in learning both the scene's appearance and geometry.
+ The experimental results look promising, and the proposed work significantly leads state-of-the-art methods.

**Weaknesses:**

- The whole pipeline seems verbose since three independent modules are stitched together with few connections. Could the proposed UC-NeRF be trained in an end-to-end manner? Additionally, the efficiency comparisons of different methods are expected to be provided in the experiments.
- The first two contributions, i.e., Layer-based Color Correction and Virtual Warping, are kind of trivial and have limited novelty. They are constructed based on existing methods like the pretrained segmentation model, the MVS model, and a geometric consistent check approach. The procedures of these two parts perform a preprocessing-like role in the proposed method. The authors are suggested to give more clarifications and highlight their specific contributions.
- For the color correction part, it seems that the accuracy of the correction performance highly depends on the sky segmentation. However, the cases shown in the paper only contain clean skies and sunny weather. I am wondering how this work performs under diverse weather conditions. Because this work aims at multi-camera systems that are widely used in outdoor scenes (such as autonomous driving), the real-world application would be preferred over the method itself.
- For the proposed Spatiotemporally Constrained Pose Refinement, please clarify its relationship and difference to the bundle adjustment.

**Questions:**

How's the time cost to filter out inaccurate depths through a geometric consistency check?

---

> ### Author Response · Authors · 2023-11-19
> **Response to Reviewer 9RAi (Part 1/2)**
>
> **1） The whole pipeline seems verbose since three independent modules are stitched together with few connections. Can the proposed UC-NeRF be trained in an end-to-end manner?**
>
> Yes, our NeRF is trained end-to-end with the learnable color correlation and virtual warping as data augmentation. The pose refinement is performed as a preprocessing step to provide accurate poses for NeRF training.
> Our method presents a unified framework for solving the problems of neural rendering with under-calibrated multi-view cameras, where the three modules are designed to tackle the key problems in this setting, such as inaccurate poses, and inconsistent color images.
> The proposed three modules can complement each other and contribute as a whole to improve the final rendering results, which are correlated with each other and effective.
>
> **2） The efficiency comparisons of different methods.**
>
> Thanks for your suggestion. We add Tab. 1 below to the paper for efficiency comparison. Note that our training time includes both steps (pose refinement and NeRF training) described in Sec. 3.5. Zip-NeRF is more efficient than other methods except Instant-NGP, which is specifically designed for NeRF acceleration. Since our method is built upon Zip-NeRF, our method consumes a bit more time than Zip-NeRF but achieves a significant improvement in rendering quality.
>
> Table 1. Efficiency Analysis. Tested on one NVIDIA Tesla V100 GPU with image resolution $1920 \times 1280$.
> | Method  | Training | Inference (per image) | PSNR |
> |-------|:-: | :-:|  :-:|
> | Mip-NeRF |20h| 70s| 22.42|
> |Mip-NeRF-360 | 14h | 42s | 24.46|
> |Instant-NGP | 30min | 0.35s |23.84|
> |S-NeRF | 15h | 80s | 24.89|
> |Zip-NeRF | 2h | 2s | 26.21|
> |Ours | 3h | 3.2s | 28.13 |
>
> **3） The highlight of the first two contributions.**
>
> The first two modules are embedded in the NeRF framework for end-to-end training. They respectively address a specific challenge of color correction during training of NeRF in multi-camera systems, i.e., **(i)** the way to model color correction and **(ii)** color correction in limited observations.
>
> **(i)** Existing methods do not perform well in color correction in our setting, as shown in Fig. 11 and Tab. 5 (in Appendix A.3.2) of our revised paper. They generally alleviate the inconsistent color supervision by modeling image-dependent appearance with a global latent code for each image. However, the lack of explicit modeling for color correction from the latent code can lead to the disentanglement of attributes unrelated to color inconsistency. Moreover, the capacity of a global latent code to uniformly correct colors in different regions of an image is limited, especially when different regions correspond to different color transformations. We propose the concept of layer-wise color correction.
> We approximate color corrections in foreground and sky regions as different affine transformations. This strategy not only avoids the disentanglement of attributes unrelated to color inconsistency when the latent code implicitly models the image-dependent appearance, but also ensures the ability to model different color transformations in different regions of a single image.
>
> **(ii)** The limited observations used for color correction. Learning one color correction for each image leads to overfitting when the training images lack color and viewpoint diversity, especially for areas observed by side-view cameras, which have fewer observations and limited overlapping with front-view areas. Virtual warping provides a novel way to improve color correction. It provides diverse color observations from more viewpoints of different cameras for each 3D region, alleviating the overfitting of color correction and improving the 3D recovery.
>
> **4) The performance under diverse weather conditions.**
>
> Thanks for your suggestion. Our datasets have included sunny and overcast weather, and we add rainy and night conditions here. As shown in Tab. 2 below, our approach notably improves across these scenarios. Please check the rendered images in the Fig. 13 (in Appendix A.3.2) of the revised paper and the video in the supplementary material.
>
> Table 2. Comparison of diverse weather conditions. Tested on Waymo Segment-100170 (Rainy) and Waymo Segment-150908 (Night).
> | Method  | PSNR (Rainy) ↑ | SSIM (Rainy) ↑ | LPIPS (Rainy) ↓ | PSNR (Night) ↑ | SSIM (Night) ↑ | LPIPS (Night) ↓ |
> |-------|:-: | :-:|  :-:| :-: | :-:|  :-:|
> | Zip-NeRF |27.65 | 0.831 | 0.434 | 30.69 | 0.864 | 0.512 |
> | UC-NeRF (Ours) | **30.03** | **0.866** | **0.387** | **31.32** | **0.869** | **0.491**|

---

> ### Author Response · Authors · 2023-11-19
> **Response to Reviewer 9RAi (Part 2/2)**
>
> **5) The relationship and difference to the bundle adjustment.**
>
> Bundle adjustment (BA) is a widely used technique that involves simultaneously adjusting pose parameters and 3D point locations.
> In our framework, we perform BA to optimize poses, and additionally incorporate the spatiotemporal constraint between cameras to eliminate errors of the under-constrained pose estimation of small-overlapping cameras in a multi-camera system.
>
> **6) How much does the time cost to filter out inaccurate depths through a geometric consistency check?**
>
> The geometric consistency check takes about 0.3 seconds per depth map. Thus, for a 300-image scene, it could be done in 90 seconds.

---

> > ### Comment · Reviewer_9RAi · 2023-11-22
> > **Thanks for the Response**
> >
> > Thanks for your response. It addressed some of my concerns. However, I am still confused about the relationship between the proposed Spatiotemporally Constrained Pose Refinement and BA. In your reply, did you mean jointly performing BA and the proposed spatiotemporal constraint? Such an implementation may be less significant to the contribution. More details are expected to be discussed here.

---

> ### Author Response · Authors · 2023-11-22
> **Response to Reviewer 9RAi about BA and the proposed spatiotemporal constraint**
>
> Thanks for your feedback. We greatly appreciate your questions and suggestions concerning our pose optimization module. First, we would like to emphasize that our UC-NeRF is designed to address the problem of color inconsistency in NeRF training under a multi-camera system, and pose refinement is only a part of this. It works in conjunction with other modules to tackle the challenges of color inconsistency.
>
> In our spatiotemporally constrained pose refinement module, we optimize the poses using bundle adjustment (BA). However, BA is a general pose optimization technique that relies on different energy functions to optimize poses. Many works enhance pose accuracy through the design of energy functions, such as Pixel Perfect SFM[1] rewrites reprojection loss as pixel's feature loss, and NIM-SLAM [2] rewrites pixel's photometric loss as multi-scale patch’s photometric loss. For our multi-camera setting, we also redesign the energy function. We consider the prior of the cameras, i.e., their spatiotemporal relationships, and add additional constraints to the poses. We design the energy function in the form of Eq.1 below:
>
> $$
> L=\sum_{((i, k),(j, l)) \in \mathcal{E}}\left\|\mathbf{p}_l^j-\Pi_l\left(\left(\mathbf{T}^j \Delta \mathbf{T}_l\right)^{-1} \mathbf{T}^i \Delta \mathbf{T}_k \Pi_k^{-1}\left(\mathbf{q}_k^i\right)\right)\right\|^2. \tag{1}
> $$
>
> $\mathbf{T}^{i}_{k}$, which is the kth camera's pose at time $i$, is modeled as  $\mathbf{T}^i \Delta \mathbf{T}_k$ with two parts: the ego pose $\mathbf{T}^i$, and the temporally consistent and optimizable relative transformation of the cameras $\Delta \mathbf{T}_k$. The inclusion of this constraint significantly improves the rendering results, as shown in Tab. 1 below and Fig.12 in the appendix of the paper.
>
> Table 1. Effectiveness of spatiotemporal constraint tested on ten Waymo scenes.
> | Method  | PSNR ↑   | SSIM ↑   | LPIPS ↓ |
> |-------|-------|-------|-------|
> | W/O Spatiotemporal Constraint | 27.89 | 0.835 | 0.368|
> | W Spatiotemporal Constraint  | **28.13** | **0.842** |**0.356**|
>
> Otherwise, independently modeling camera poses without the proposed spatiotemporal constraint is feasible, but it may lead to less accurate pose estimation. This is primarily because images captured by a moving vehicle share limited overlaps, especially among the images captured by different cameras. This could potentially result in insufficient connections between images taken by different cameras, making the pose optimization relatively less constrained and unstable.
>
> In addition, we want to state the motivation for using BA. We have tried various strategies for joint optimization of NeRF and pose in multi-camera systems. However, due to the color differences between images, joint optimization of NeRF, pose, and color correction through photometric loss only might not provide sufficient constraints. Tab. 2 below (also shown as Tab. 6 in our paper) also demonstrates that explicitly establishing pixel correspondence to optimize poses shows more significant improvement than jointly learning poses, NeRF, and color correction.
>
> Table 2. Comparison of our pose refinement method with nerf-pose joint refinement methods BARF, NeRF-- (the same pose refinemment strategy in Block-NeRF). Tested on a randomly selected Waymo scene.
>
> | Method  | PSNR ↑   | SSIM ↑   | LPIPS ↓ |
> |-------|-------|-------|-------|
> |NeRF$--$ | 28.48 | 0.851 | 0.383 |
> |BARF|  28.65 | 0.856 |  0.373 |
> |UC-NeRF (Ours) | **29.14** | **0.867** | **0.355** |
> ---
> [1] Lindenberger P, Sarlin P E, Larsson V, et al. Pixel-perfect structure-from-motion with featuremetric refinement[C]//Proceedings of the IEEE/CVF international conference on computer vision (ICCV). 2021: 5987-5997.
>
> [2] Li H, Gu X, Yuan W, et al. DENSE RGB SLAM WITH NEURAL IMPLICIT MAPS[C]//The Eleventh International Conference on Learning Representations (ICLR). 2022.

---

> > ### Comment · Reviewer_9RAi · 2023-11-22
> > **Thanks**
> >
> > Thanks for the detailed reply and insightful analysis. I would like to increase my original rating.

---

> > > ### Author Response · Authors · 2023-11-22
> > > **Thanks for your discussion**
> > >
> > > Thanks for your suggestions and discussions about our work. This also helps others better understand the motivation behind each of our modules. Lastly, we are very grateful for your support of our work.

---

### Official Review · Reviewer_oaML · 2023-11-06

**Soundness:** 3 good
**Presentation:** 3 good
**Contribution:** 3 good
**Rating:** 6
**Confidence:** 4

**Summary:**

The paper proposes a neural rendering system for automotive multi-camera temporally captured data (which they should explicitly mention in title and is misleading if not mentioned), which accounts for color variation, extrinsic errors and lack of sufficient training data typically effecting previous automotive applied Nerf methods. They first handle extrinsic errors by SLAM and again do something similar to SLAM as post-processing to recalibrate the cameras across time. To handle color variation across cameras at a given time instant, an affine color correction matrix is learnt separately from sky and foreground (as they can get sky mask from previous work) and thus there are separate NeRF models. Since NeRF requires dense sampling of input images, they also propose to get novel views via a separate pre-existing MVS method and use that depth map to render novel views. These novel views are then added to the training set of images. A new NeRF optimizing function is then defined taking all these into account. The results show improvement compared to many existing SOTA methods.

**Strengths:**

The paper handles a relevant problem in real life data. The proposed method looks sound. The references look adequate. The comparative results look good.

**Weaknesses:**

- First, I would like the title to be more specific. Its very misleading as the paper does use automotive setting (sky+fg) as necessary input. The title seems to suggest that its a generic method for uncalibrated cameras. Also all results are on automotive. Please correct it.

- In Eq. 5, what is b and d. I think the authors missed defining those params.

- In Eq.7, isn't it  better to reduce reprojection error as a function of direct 4x4 transformation between the cameras across time. For example directly modeling the transformation between cameras labeled T^i*delta_T1 and T^j * delta_T1 in Fig4. The reason for this is that at time t=0, its convenient to bring all the three cameras in Fig4 to the car's coordinate system, but later assuming that this transformation is fixed and won't perturb due to camera shake, bad roads, bumps etc. is an unrealistic assumption. Then propagating this incorrect assumption across time from T^i to T^j can lead to erroneous extrinsics estimation in Eq. 7.

- When virtual views are created and added to the training set as discussed in Section 3.3, it has holes either due to low confidence or occlusion as shown in Fig 9, then how does the sky segmentation from Yin 2022 perform in these images. What happens to the mask value in the missing image regions in virtual view and how does in impact Equation 3.

- The training strategy is not very clear. So, you train using Eq.4 upto some convergence, Then you get A and C color correction matrices. Then again you apply the corrected A and C matrices to virtual views in Section 3.3. Then you get color corrected virutal views. Then you again train the original set of spatial+temporal data but this time include virtual views? I think Section 3.5 needs more explanation because it joins all your individual modules and is critical to understand how the complete system is working.

- Section 4.4 is redundant I think. It has nothing to do with the main goal of your paper and that space could have been used to explain your main parts e.g. Section 3.5 in detail.


- In Fig5, the part of the image where the road appears to merge, the green region adjoining the bright sky appears to be hazy in the proposed result compared to Zip-Nerf results. In other words Zip-Nerf results are much sharper in that region. What could be the reason for that?

**Questions:**

Kindly address the weakness as much as possible. I will update my review based on the rebuttal. Currently its borderline for me.

---

> ### Author Response · Authors · 2023-11-19
> **Response to Reviewer oaML**
>
> **1) The title needs to be more specific.**
>
> Thanks for your suggestion. Our method aims at novel view rendering with under-calibrated multi-camera systems, where autonomous driving is one of the typical scenarios. Moreover, public autonomous driving datasets like Waymo are widely used benchmarks, making it convincing to compare our method with other works using the datasets. We are also open to incorporating ''autonomous driving'' in the title if needed.
>
> **2) The missing definition of ‘b’ and ‘d’.**
>
> Thanks for pointing this out. **b** and **d** should be **x** and **y** as the translation term in the color correction matrix. Please check our general response 2 about the typos and symbols.
>
> **3) Isn't it better to reduce reprojection error as a function of direct 4x4 transformation between the cameras across time?**
>
> Thanks for your suggestion. As you pointed out, the relative pose between cameras may change during driving, but such changes are typically caused by accumulated shakes over a vehicle's long-term driving like several months. NeRF models generally operate on very short video sequences lasting from minutes to hours, during which the changes among the relative poses of the multiple cameras are subtle. On the contrary, directly modeling the transformation between cameras without the proposed spatiotemporal constraint is feasible, but it may lead to less accurate pose estimation. This is primarily due to the fact that images captured by a moving vehicle share limited overlaps, especially among the images captured by different cameras. This could potentially result in insufficient connections between images taken by different cameras, making the pose optimization relatively less constrained and unstable.
> Considering both factors, we propose to leverage the spatiotemporal constraint to enhance the pose refinement.
>
> We add an experiment in Tab. 1 below to analyze the spatiotemporal constraint. The experiment is conducted in ten commonly used self-driving sequences without any special selection. Compared to directly modeling the transformation without any constraint, our spatiotemporal constrained pose refinement leads to higher rendering quality of the reconstructed NeRF with more accurate camera poses. Fig. 12 of our revised paper also illustrates the improvement of rendering from this constraint.
>
> Table 1. Effectiveness of spatiotemporal constraint tested on ten Waymo scenes.
> | Method  | PSNR ↑   | SSIM ↑   | LPIPS ↓ |
> |-------|-------|-------|-------|
> | W/O Spatiotemporal Constraint | 27.89 | 0.835 | 0.368|
> | W Spatiotemporal Constraint  | **28.13** | **0.842** |**0.356**|
>
> **4) How does the segmentation model perform in virtual views?**
>
> We do not employ any segmentation in virtual views. Both the sky mask and color of a pixel in the virtual view are obtained from the corresponding pixel in the original frames of the input sequence based on the warping by depth.
>
> **5) The training strategy is not very clear. Sec. 4.4 is redundant. Explain Sec. 3.5 in detail.**
>
> Thanks for your suggestion. We revise the training strategy in Sec. 3.5 (detailed in general response 1) and move Sec. 4.4 to the appendix.
>
> **6) The green region adjoining the bright sky appears to be hazy.**
>
> Thanks for the observation. The observed blurriness may stem from some random factors, such as sky segmentation errors or insufficient supervision of the regions due to few samplings. We have noticed similar blurriness in the rendering results of Zip-NeRF, whereas our results are clearer, as shown in Fig. 13 of our revised paper. On the whole, our rendering quality considerably outperforms that of Zip-NeRF in numerous areas, like consistent colors, sharp details, and clear logos.

---

> > ### Comment · Reviewer_oaML · 2023-11-22
> >
> > Thanks for the rebuttal. I am satisfied with the responses and paper changes except for the response for title change. The paper explicitly uses sky mask which is very specific to autonomous related works, while the current paper title seems to reflect that the solutions for uncalibrated is a generic solution. I think the current title is quite misleading and definitely changed. In rebuttal authors say "...where autonomous driving is one of the typical scenarios...", but this is THE only scenario shown in paper and the proposed pipeline does use the specific information of sky and foreground objects.

---

> ### Author Response · Authors · 2023-11-22
> **Response to Reviewer oaML about the title**
>
> Thanks for your valuable feedback. We greatly appreciate your suggestion regarding the title and agree with it. Accordingly, we have incorporated the concept of 'autonomous driving' into the title in the most recent revision of our paper.

---

### Author Response · Authors · 2023-11-19
**General response to all reviewers**

We are grateful to all reviewers for their insightful and constructive suggestions.
We are glad that reviewers found: (1) The work is sound / technically sound (Reviewer oaML, 5Tdv, ehJ8) and effective (Reviewer ehJ8); (2) The problem setting is practical (Reviewer oaML) and interesting (Reviewer ehJ8); (3) The comparative results look good/promising (Reviewer oaML, 9RAi). We have revised our paper accordingly, marking changes with blue text for easy identification.
Each comment from the reviewers has been replied individually. Here we provide a summary of their common problems.

**1) Training strategy**

Our training strategy mainly includes two parts: **(i)** Pose refinement and depth estimation.
We initialize the poses from sensor-fusion SLAM and further optimize them using our proposed spatiotemporally constrained pose refinement module, as described in Eq. 7 of the paper. With these refined poses, we generate a depth map and geometric consistency mask for each image, following the procedure outlined in Sec. 3.3 of the paper. **(ii)** End-to-end NeRF optimization. Specifically, the proposed layer-based color correction and virtual warping are used in the optimization of NeRF to achieve high-quality renderings.
In each training batch, we randomly sample $B$ real images, and employ our virtual warping module to create $V$ virtual views for each real image. The pixels are randomly sampled from these real and virtual views as the ground truth for NeRF training. Our UCNeRF renders these pixels based on Eq. 4 of the paper and is supervised by the loss function in Eq.1 below:

$$
L=L_{p h o}+\lambda L_{\text {sky }}+\gamma L_{\text {reg}}, \tag{1}
$$

where $\lambda$ and $\gamma$ are the weights of $L_{\text {sky }}$ and $L_{\text {reg}}$.


**2) Typos and symbols**

In Eq. 5, $\mathbf{b}$ and $\mathbf{d}$ should be $\mathbf{x}$ and $\mathbf{y}$ respectively, which represent the translation term of the color correction matrix.

In Eq. 6, $d_v$ and $d_o$ refer to the pixel depth in virtual views and the corresponding pixel depth in original real views.
All required definitions have been added to the paper.

---

### Meta-Review · Area_Chair_EEta · 2023-12-05

**Metareview:**

This paper presents a method for new view image synthesis in multi-camera systems tailored for autonomous driving scenarios. The novelty of the work lies in layer-based color correction for addressing color inconsistencies in training images. It also incorporates a virtual-warping strategy for taking advantage of the multi-camera setting. The experimental results show superior performance than state-of-the-art methods.

The major strengths of the paper are:
(1) New layer-based color correction and virtual warping strategy to improve the novel view synthesis.
(2) A new spatiotemporally constrained pose refinement is effective.
(3) The result is strong.

On the other hand, the chief weakness was that the paper was somewhat misleading. Although the method is specifically designed for a multi-camera system that was moving outdoors (where the sky is observed), it was originally described in a general multi-camera setting. The paper has been corrected to some extent during the author-reviewer discussion period, but the paper still has the original flavor and is misleading.

Overall, the ratings are positive except for one negative rating. The paper was on the borderline and was discussed among AC and reviewers. We consider that the paper has merit in that the proposed strategies clearly show effectiveness, which can be further applied in other contexts by relaxing some of the restricting assumptions made in the paper; therefore, worth sharing. The reviewers and AC read the rebuttal and took it into consideration for the final recommendation.

**Justification For Why Not Higher Score:**

Although we see the novelty in the paper, i.e., layer-based color correction, virtual warping technique, and spatiotemporally constrained pose optimization, each of them makes a moderate but not significant contribution. They are tailored for a specific setting that the paper is considering, and therefore it is natural to incorporate these techniques.

**Justification For Why Not Lower Score:**

We appreciate the quality of the result. The results show the method's superiority compared to the state-of-the-art methods.

---

### Decision · Program_Chairs · 2024-01-16

Accept (poster)